# FedSSM: State Space Model-based Proactive Inference for Heterogeneous Multimodal Federated Learning

Hengyi Ren[1]  Yuchen Xie[1]  Changlong Wang[2]  Xin Li[* 3]  Yue Huang[4]  Jian Guo[4]  Lijuan Sun[4]

## Abstract

Multimodal Federated Learning (MMFL) addresses collaborative training across clients with heterogeneous modality configurations, where effective client selection becomes critical under the compounded challenges of modality, distribution, and quantity heterogeneity. Existing selection methods operate within a reactive paradigm, responding to current observations without anticipating how decisions influence future optimization trajectories. This myopic approach leads to suboptimal convergence when training dynamics shift rapidly under severe heterogeneity. We propose FedSSM, which reconceptualizes client selection as a proactive decision-making process by predicting training dynamics through decision-aware state space models. The prediction error yields a *surprise* signal that quantifies uncertainty and governs adaptive participation budgets and exploration-exploitation trade-offs via counterfactual reasoning over candidate actions. For aggregation, we introduce trust-weighted fusion with modality-specific routing, where surprise calibrates sensitivity to client anomalies. Experiments on four multimodal benchmarks demonstrate that FedSSM achieves 2.5–4.5% accuracy improvements over state-of-the-art methods while reducing communication rounds by over 30%.

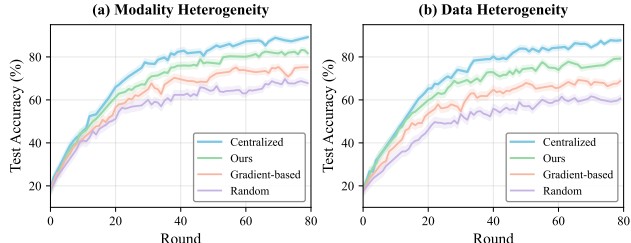

*Figure 1.* Convergence comparison under modality and data heterogeneity on UCF-101 with 100 clients, Dirichlet $\alpha = 0.5$, and 50% partial-modality ratio. We FedSSM against Power-of-Choice (Cho et al., 2022) as a representative gradient-based selection method, FedAvg (McMahan et al., 2017) with random selection, and a centralized training oracle with full data access.

## 1. Introduction

The proliferation of multi-sensory devices has catalyzed an unprecedented surge in multimodal data generation across diverse domains, from autonomous vehicles integrating visual, LiDAR, and audio streams to healthcare systems combining imaging and physiological signals (Feng et al., 2023; Xiong et al., 2022). Such data, inherently distributed and privacy-sensitive, cannot be centralized for conventional training without raising significant regulatory concerns. Federated learning (McMahan et al., 2017) has emerged as a principled paradigm enabling collaborative model training while preserving data locality, yet extending this framework to multimodal scenarios introduces challenges that transcend those encountered in unimodal settings (Chen & Zhang, 2022; 2024).

Multimodal federated learning must contend with a tripartite heterogeneity structure that fundamentally complicates the optimization landscape. Clients equipped with different sensor suites possess varying subsets of modalities, creating architectural mismatches that preclude straightforward parameter aggregation (Zhang et al., 2024). Beyond this modality heterogeneity, the statistical properties of local data distributions diverge considerably across clients (Li et al., 2020; Zhao et al., 2018), while quantity heterogeneity arising from vastly different sample sizes further biases aggregation toward data-rich participants. The interplay among these three dimensions engenders optimization dy-

---
[1]College of Information Science and Technology & Artificial Intelligence, Nanjing Forestry University, Nanjing, 210000, China [2]College of Computer Science, Nanjing Audit University, Nanjing, 210000, China [3]College of Computer Science and Software Engineering, Hohai University, Nanjing, 210000, China [4]School of Computer Science, Nanjing University of Posts and Telecommunications, Nanjing, 210000, China. Correspondence to: Xin Li <li-xin@hhu.edu.cn>.

*Proceedings of the $43^{rd}$ International Conference on Machine Learning*, Seoul, South Korea. PMLR 306, 2026. Copyright 2026 by the author(s).

namics far more volatile than those observed when any single heterogeneity type dominates.

A natural question arises: how should the central server orchestrate client participation under such compounded heterogeneity? Recent years have witnessed substantial progress in client selection strategies, including utility-based methods (Lai et al., 2021), gradient-based approaches (Cho et al., 2022; Wang et al., 2023), and learning-based frameworks (Wang et al., 2020; Yan et al., 2023). Despite this methodological diversity, existing approaches share a fundamental limitation: they lack an explicit model of how selection decisions causally influence optimization trajectories. Utility-based frameworks prioritize greedy maximization of current-round contributions while neglecting multi-round consequences. Gradient-based strategies rely on retrospective metrics that serve as lagging indicators of past training rather than predictors of future dynamics. While reinforcement learning approaches such as FAVOR (Wang et al., 2020) implicitly optimize for long-term rewards through policy learning, they remain *model-free* in that they do not construct an explicit predictive model of training dynamics. This absence of an explicit world model limits their ability to perform direct counterfactual reasoning about how alternative selection strategies would affect convergence.

We argue that effective client orchestration demands a *proactive* approach capable of predicting training dynamics and reasoning about consequences before committing to decisions. The key insight is that client selection in federated learning is fundamentally a sequential decision problem: actions at round $t$ influence observations at round $t + 1$, which in turn affect future decisions. This causal structure suggests that effective selection requires modeling how different strategies lead to different optimization trajectories—a capability that reactive methods inherently lack.

To realize this vision, we propose FedSSM, which integrates state space models (Gu et al., 2021; 2020) for proactive client selection. The core idea is to maintain a predictive model that captures temporal dependencies in training dynamics and forecasts future states given current observations and actions. At the heart of FedSSM lies a *decision-aware* state space model where transition matrices are modulated by participation decisions, enabling the model to learn how different selection strategies causally influence convergence. The discrepancy between predictions and observations yields a *surprise* signal that quantifies prediction uncertainty: high surprise indicates unfamiliar optimization terrain requiring more information acquisition, while low surprise suggests predictable dynamics where resources can be conserved. This surprise signal governs both adaptive participation budgets and exploration-exploitation trade-offs through counterfactual reasoning over candidate actions. For aggregation, we introduce trust-weighted fusion with

modality-specific routing, where surprise calibrates sensitivity to client anomalies.

The main contributions of this work are: **(1)** We identify the fundamental limitation of model-free client selection under volatile multimodal heterogeneity and formalize the distinction between model-free and model-based selection paradigms. **(2)** We propose FedSSM, a novel framework featuring decision-aware state space modeling that predicts training dynamics, surprise-driven adaptive budgeting, and counterfactual client selection that reasons about alternative participation strategies. **(3)** We conduct extensive experiments on four multimodal benchmarks, demonstrating that FedSSM achieves 2.5–4.5% accuracy improvements over state-of-the-art methods while reducing communication rounds by over 30%.

## 2. Related Work

### 2.1. Multimodal Federated Learning

Multimodal federated learning addresses training across clients with heterogeneous modality configurations. Early works employed modular architectures with blockwise aggregation (Chen & Zhang, 2022; Xiong et al., 2022), while recent methods tackle broader heterogeneity: FedM-Bridge (Chen & Zhang, 2024) uses topology-aware hypernetworks for diverse fusion strategies, and FIMCFG (Chao et al., 2025) handles incomplete multi-view scenarios through graph guidance. For client selection, strategies have evolved from uniform sampling (McMahan et al., 2017) to utility-based (Lai et al., 2021), gradient-based (Cho et al., 2022), and learning-based methods (Yan et al., 2023; Wang et al., 2020). Utility-based and gradient-based approaches select clients based on observed metrics without modeling future consequences. Learning-based methods such as FA-VOR (Wang et al., 2020) employ reinforcement learning to optimize long-term objectives, yet they remain model-free and do not explicitly predict how different selection strategies affect training dynamics. Our work departs from these approaches by treating the server as a model-based agent that constructs an explicit predictive model of optimization trajectories, enabling direct counterfactual reasoning over candidate actions.

### 2.2. State Space Models for Sequential Decision Making

State space models (SSMs) have emerged as efficient alternatives to Transformers for long-sequence modeling. The S4 architecture (Gu et al., 2021) achieves linear-time complexity through continuous-time dynamics, while HiPPO (Gu et al., 2020) provides principled compression of historical information via optimal polynomial projections. Mamba (Gu & Dao, 2024) further introduces selective state spaces where parameters become input-dependent. These properties make

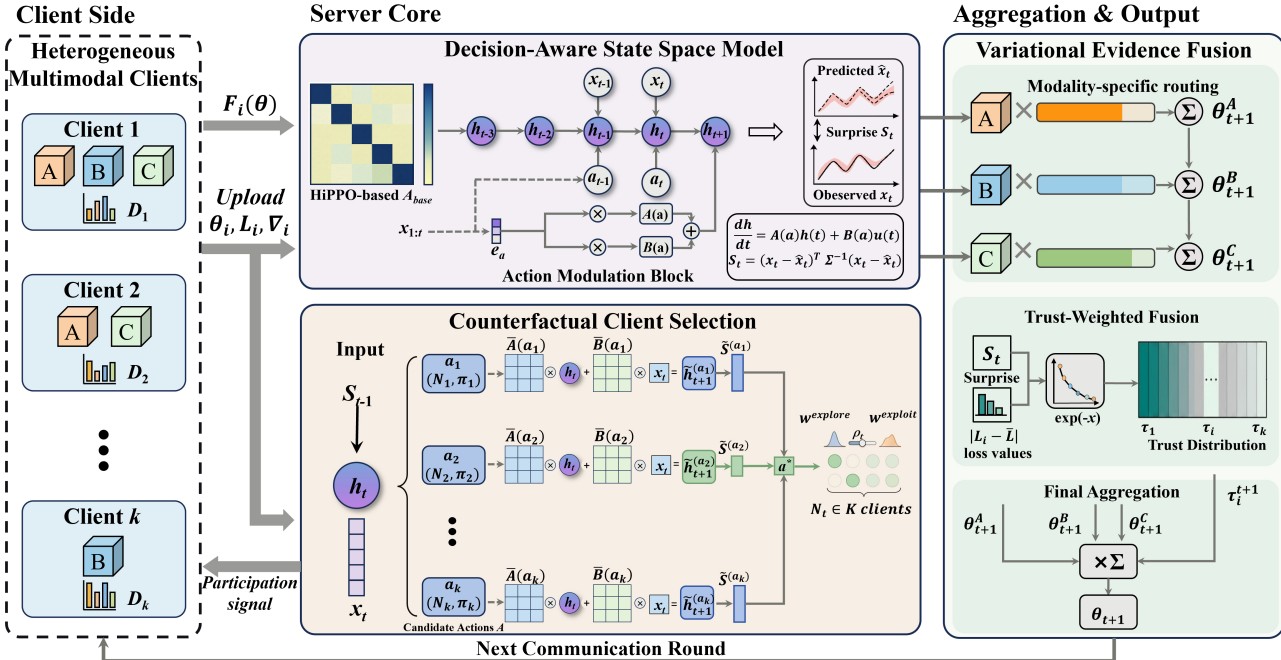

*Figure 2.* Overview of the FedSSM framework. **Left:** Clients participate with heterogeneous modality configurations, where blocks **A**, **B**, and **C** denote distinct modalities (*e.g.*, image, audio, text). **Center:** The server employs a Decision-Aware State Space Model to predict training dynamics and perform counterfactual client selection. **Right:** Global aggregation utilizes Variational Evidence Fusion, which applies modality-specific routing and trust weights derived from the surprise signal to integrate local updates.

SSMs naturally suited for federated optimization, which spans hundreds of rounds with complex temporal dependencies. However, existing SSM applications focus on passive sequence modeling rather than *active* decision making. We extend SSMs by introducing action-conditioned dynamics, enabling the model to capture how different selection strategies causally influence convergence—a capability essential for proactive client orchestration.

## 3. Methodology

### 3.1. Problem Formulation

Consider a federated system comprising $K$ clients coordinated by a central server. Let $\mathcal{M} = \{1, \dots, M\}$ denote the set of modality types. Each client $i \in [K]$ possesses a modality subset $\mathcal{M}_i \subseteq \mathcal{M}$, with *full-modality clients* $\mathcal{K}^{\text{full}} = \{i : \mathcal{M}_i = \mathcal{M}\}$ and *partial-modality clients* $\mathcal{K}^{\text{part}} = \{i : \mathcal{M}_i \subsetneq \mathcal{M}\}$. Each client $i$ holds a private dataset $\mathcal{D}_i$ with $n_i$ samples.

The server maintains a global model $f(\cdot; \boldsymbol{\theta})$ with $\boldsymbol{\theta} = \{\boldsymbol{\theta}^1, \dots, \boldsymbol{\theta}^M, \boldsymbol{\theta}^{\text{fus}}\}$. The global objective is:

$$\min_{\boldsymbol{\theta}} \mathcal{F}(\boldsymbol{\theta}) = \sum_{i=1}^{K} p_i F_i(\boldsymbol{\theta}; \mathcal{D}_i), \qquad (1)$$

where $F_i(\boldsymbol{\theta}; \mathcal{D}_i) = \frac{1}{n_i} \sum_{(\mathbf{x}, y) \in \mathcal{D}_i} \ell(f(\mathbf{x}; \boldsymbol{\theta}), y)$ is the local

empirical risk and $p_i = n_i / \sum_j n_j$ is the sample-weighted contribution.

The heterogeneity manifests in three dimensions: (1) *modality heterogeneity*: $|\mathcal{M}_i| \in \{1, \dots, M\}$; (2) *distribution heterogeneity*: $P_i(\mathbf{x}, y) \neq P_j(\mathbf{x}, y)$ for $i \neq j$; and (3) *quantity heterogeneity*: $n_i$ differs significantly. These three factors are coupled through the selected client set. Local training, gradient estimation, and global aggregation are all performed only on the clients selected in the current round. Therefore, client selection is not only a communication-saving mechanism, but also the entry point through which missing modalities, skewed labels, and imbalanced data quantities affect the global update.

At each round $t$, the server determines the action $\mathbf{a}_t = (N_t, \boldsymbol{\pi}_t)$, comprising the participation budget $N_t \in [N_{\min}, N_{\max}]$ and selection distribution $\boldsymbol{\pi}_t \in \Delta^K$. A desirable selection policy should decide both how many clients are needed and which clients are informative under the current training state. This is difficult in MMFL because the same client can be useful in one stage but harmful in another stage, especially when modality availability and data distributions change the optimization trajectory over time.

**Definition 3.1** (Model-Free Client Selection). A selection strategy is *model-free* if it determines $\boldsymbol{\pi}_t$ based solely on past observations without explicitly modeling future dynam-

ics:

$$\boldsymbol{\pi}_t = g(\mathbf{x}_{t-1}, \mathbf{x}_{t-2}, \ldots, \mathbf{x}_1), \tag{2}$$

where $\mathbf{x}_\tau$ denotes observed training metrics at round $\tau$. Examples include gradient-based selection (Cho et al., 2022) and utility-based methods (Lai et al., 2021).

**Definition 3.2** (Model-Based Client Selection). A selection strategy is *model-based* if it determines the action by evaluating predicted future trajectories over a candidate action space $\mathcal{A}$:

$$\mathbf{a}_t = \arg\max_{\mathbf{a}\in\mathcal{A}} \mathcal{U}\big(\hat{\mathbf{x}}_{t+1}(\mathbf{a}), \hat{\mathbf{x}}_{t+2}(\mathbf{a}), \ldots \mid \mathbf{x}_{1:t-1}\big), \tag{3}$$

where $\mathcal{A}$ is a finite set of candidate actions, $\hat{\mathbf{x}}_\tau(\mathbf{a})$ is the predicted future state conditioned on candidate action $\mathbf{a}$, and $\mathcal{U}(\cdot)$ is a utility function evaluating predicted trajectories.

The key distinction is that model-free methods respond to *what has happened*, while model-based methods anticipate *what may happen* by comparing predicted consequences of candidate actions. This difference is important in heterogeneous MMFL. A bandit-style selector updates action scores from realized rewards, but it does not explicitly predict how the next loss, gradient variance, or modality composition will change after a candidate action. FedSSM instead builds a lightweight dynamics model of the training process, allowing the server to compare candidate decisions before execution.

### 3.2. Main Idea of FedSSM

**Rethinking Client Selection as Sequential Decision Making.** Client selection in federated learning is fundamentally a sequential decision problem: actions at round $t$ influence observations at round $t + 1$, which in turn affect future decisions. This causal structure suggests that effective selection requires modeling the decision-observation dynamics, i.e., how different selection strategies lead to different optimization trajectories.

In multimodal FL, this sequential effect is more pronounced. Selecting many partial-modality clients may reduce communication or improve coverage for one modality, but it can also increase fusion instability in later rounds. Selecting clients with large local datasets may improve statistical efficiency, but it may also introduce stragglers or amplify distribution bias. Thus, the selection policy should not only rank clients by their past utility, but also estimate how the current selection will change the future training state.

**Definition 3.3** (Predictive Selection Function). We introduce a predictive selection function $h(\cdot; \phi)$ parameterized by $\phi$, which maps the current state and candidate actions to predicted future outcomes:

$$\boldsymbol{\pi}_t, N_t := h(\mathbf{h}_t, \mathcal{A}; \phi), \tag{4}$$

where $\mathbf{h}_t \in \mathbb{R}^d$ is a latent state summarizing the optimization history, and $\mathcal{A}$ is the action space. The function $h$ serves as a predictive model enabling the server to reason about counterfactual consequences before committing to decisions.

The key insight is that instead of directly mapping observations to actions, we introduce an intermediate latent state $\mathbf{h}_t$ that compresses historical information and supports forward prediction. This parallels how FedMBridge (Chen & Zhang, 2024) introduces a bridge function to handle architecture heterogeneity; we introduce a *predictive function* to handle temporal heterogeneity in training dynamics. The predictive function is not intended to replace existing FL optimizers. Rather, it operates on the server side and guides which clients should participate in the next round.

**Framework Overview.** FedSSM consists of three components:

1. **Decision-Aware State Space Model** (§3.3): Maintains latent state $\mathbf{h}_t$ and predicts future observations $\hat{\mathbf{x}}_{t+1}$ conditioned on actions.

2. **Counterfactual Client Selection** (§3.4): Evaluates candidate actions by simulating their consequences through the state space model.

3. **Trust-Weighted Aggregation** (§3.5): Uses prediction uncertainty to calibrate aggregation weights for heterogeneous client updates.

Figure 2 illustrates the overall architecture. The central signal connecting the three components is the prediction error, or *surprise*. It controls the participation budget, the exploration-exploitation balance, and the trust assigned to fusion updates. This design avoids introducing separate heuristics for each decision.

### 3.3. Decision-Aware State Space Model

The first challenge is modeling the relationship between actions and future observations. We require a model that (1) efficiently summarizes long optimization histories, (2) captures how different actions lead to different outcomes, and (3) quantifies prediction uncertainty.

**Why State Space Models?** We adopt continuous-time state space models (Gu et al., 2021) for three reasons. First, SSMs achieve linear-time sequence modeling, enabling efficient processing of training histories. Second, the HiPPO framework (Gu et al., 2020) provides principled compression of historical information into fixed-dimensional states. Third, and more importantly for FedSSM, the state transition formulation naturally supports action-conditioned dynamics.

This third property distinguishes FedSSM from standard online selection modules. A bandit or reinforcement-learning

selector can optimize long-term rewards, but it usually treats the training process as a black box and updates the policy after observing feedback. In contrast, FedSSM explicitly predicts the next optimization state before the round is executed. The SSM therefore acts as a compact world model of the FL process. It does not need to model every client in detail; it only tracks aggregate training dynamics that are most relevant to server-side decision making.

**Action-Conditioned State Transition.** The latent state $\mathbf{h}_t \in \mathbb{R}^d$ evolves according to the discrete-time dynamics:

$$\mathbf{h}_t = \bar{\mathbf{A}}(\mathbf{a}_{t-1})\mathbf{h}_{t-1} + \bar{\mathbf{B}}(\mathbf{a}_{t-1})\mathbf{x}_{t-1}, \quad \hat{\mathbf{x}}_t = \mathbf{C}\mathbf{h}_t, \quad (5)$$

where the observation vector $\mathbf{x}_t = [L_t, \|\nabla L_t\|, \sigma_t^2, t/T]^\top$ contains global loss, gradient norm, gradient variance, and normalized progress. These four statistics have clear roles: $L_t$ describes convergence progress, $\|\nabla L_t\|$ reflects optimization strength, $\sigma_t^2$ captures inter-client disagreement, and $t/T$ encodes the training stage. We use this compact state instead of high-dimensional client features to keep the server-side model lightweight and stable.

The transition matrices $\bar{\mathbf{A}}(\mathbf{a})$ and $\bar{\mathbf{B}}(\mathbf{a})$ are modulated by the action:

$$\bar{\mathbf{A}}(\mathbf{a}) = \exp(\Delta\mathbf{A}_{\text{base}}\odot\sigma(\mathbf{W}_A\mathbf{e}^a)), \quad \bar{\mathbf{B}}(\mathbf{a}) = \mathbf{B}_{\text{base}}+\mathbf{W}_B\mathbf{e}^a, \quad (6)$$

where $\mathbf{e}^a = \mathbf{W}_N\text{PE}(N_{t-1}) + \mathbf{W}_\pi\boldsymbol{\pi}_{t-1}$ embeds the action into continuous space, and $\mathbf{A}_{\text{base}}$ is initialized using HiPPO. Intuitively, selecting more clients changes the amount of information entering a round, while selecting high-variance or stale clients may perturb the trajectory more strongly. The action-conditioned transition lets the model represent these differences directly.

**Surprise as Prediction Uncertainty.** The discrepancy between prediction and observation provides a natural uncertainty signal, which we term *surprise*:

$$S_t = (\mathbf{x}_t - \hat{\mathbf{x}}_t)^\top \boldsymbol{\Sigma}^{-1}(\mathbf{x}_t - \hat{\mathbf{x}}_t), \quad (7)$$

where $\boldsymbol{\Sigma}$ is the covariance matrix estimated online via exponential moving average. High $S_t$ indicates unfamiliar optimization terrain requiring more information acquisition, while low $S_t$ suggests predictable dynamics where resources can be conserved. Since $\mathbf{h}_t$ tracks aggregate metrics rather than individual client identities, occasional client churn or stale statistics affect the surprise score only through their impact on the global training state.

### 3.4. Counterfactual Client Selection

Given the decision-aware SSM, we perform proactive selection by reasoning about counterfactual consequences of different actions.

**Counterfactual Rollout.** For any candidate action $\mathbf{a} \in \mathcal{A}$, we simulate the future state through the SSM:

$$\tilde{\mathbf{h}}_{t+1}^{(\mathbf{a})} = \bar{\mathbf{A}}(\mathbf{a})\mathbf{h}_t + \bar{\mathbf{B}}(\mathbf{a})\hat{\mathbf{x}}_t, \quad \tilde{\mathbf{x}}_{t+1}^{(\mathbf{a})} = \mathbf{C}\tilde{\mathbf{h}}_{t+1}^{(\mathbf{a})}, \quad (8)$$

where $\hat{\mathbf{x}}_t = \mathbf{C}\mathbf{h}_t$ is the current state prediction from Eq. (5). This step compares a small set of candidate budgets and exploration levels, rather than enumerating all possible client subsets. Hence, the counterfactual evaluation operates on the latent state and remains inexpensive even when the number of clients grows.

The candidate action is evaluated by:

$$\mathcal{U}(\mathbf{a}) = -\tilde{L}_{t+1}^{(\mathbf{a})} - \lambda_{\text{comm}} \cdot N, \quad (9)$$

where $\tilde{L}_{t+1}^{(\mathbf{a})}$ is the predicted loss from $\tilde{\mathbf{x}}_{t+1}^{(\mathbf{a})}$, $N$ is the participation budget in action $\mathbf{a}$, and $\lambda_{\text{comm}}$ controls communication penalty. The optimal action maximizes this utility, balancing convergence progress against communication cost.

In principle, one could enumerate all candidate actions and select the maximizer of $\mathcal{U}(\mathbf{a})$. However, the structure of the SSM and utility function admits efficient closed-form approximations. The adaptive budget and exploration-exploitation mechanisms below can be viewed as practical approximations that preserve the main trade-off without adding heavy server-side computation. Detailed scalability analysis is provided in Appendix C.

**Adaptive Participation Budget.** Rather than fixed budgets, we adapt participation based on prediction uncertainty:

$$N_t = N_{\text{min}} + \left\lfloor \frac{N_{\text{max}} - N_{\text{min}}}{1 + \exp(-\kappa(S_{t-1} - \theta_S))} \right\rfloor, \quad (10)$$

where $\theta_S$ is an adaptive threshold. When $S_{t-1}$ is high, we increase participation to gather information and stabilize the next update. When $S_{t-1}$ is low, we reduce participation to conserve communication. This makes the budget responsive to training uncertainty rather than fixed throughout training.

**Exploration-Exploitation Trade-off.** For client-level selection, we balance exploration and exploitation. The selection probability for client $i$ is:

$$q_i^{(t)} \propto (1 - \rho_{t-1})w_i^{\text{explore}} + \rho_{t-1}w_i^{\text{exploit}}, \quad (11)$$

where $w_i^{\text{explore}} = \exp(\beta_1 v_i/\bar{v}+\beta_2\Delta t_i/\bar{\Delta}t)$ encourages sampling clients with high loss variance $v_i$ or long staleness $\Delta t_i$, while $w_i^{\text{exploit}} = \exp(-\beta_3|L_i - \bar{L}|/\bar{L})$ favors clients whose losses are aligned with the current global state. Here $v_i, L_i$, and $\Delta t_i$ denote statistics from client $i$'s last participation.

The mixing coefficient $\rho_{t-1} = \sigma(\tau(\theta_S - S_{t-1}))$ ensures causal consistency: high past surprise triggers exploration, while low surprise enables exploitation. For clients that have

**Algorithm 1** FedSSM: Proactive Client Selection via Predictive State Space Models

---

**Require:** Clients $[K]$, datasets $\{\mathcal{D}_i\}$, initial model $\boldsymbol{\theta}_0$, rounds $T$
1: Initialize SSM parameters; $\mathbf{h}_0 \leftarrow \mathbf{0}$
2: **for** $t = 1, \ldots, T$ **do**
3:     // Proactive Selection
4:     Compute $\mathbf{h}_t, \hat{\mathbf{x}}_t$ via Eq. (5)
5:     Determine $N_t$ via Eq. (10); compute $\{q_i^{(t)}\}$ via Eq. (11)
6:     Sample $\mathcal{S}_t$ with $|\mathcal{S}_t| = N_t$
7:     // Local Training
8:     **for** client $i \in \mathcal{S}_t$ **in parallel do**
9:         $\boldsymbol{\theta}_{i,t} \leftarrow \text{LocalUpdate}(\boldsymbol{\theta}_t, \mathcal{D}_i, \mathcal{M}_i)$
10:     **end for**
11:     // Trust-Weighted Aggregation
12:     Observe $\mathbf{x}_t$; compute $S_t$ via Eq. (7)
13:     Compute $\{\tau_i\}$ via Eq. (13)
14:     Update $\boldsymbol{\theta}_{t+1}^m$ via Eq. (12); $\boldsymbol{\theta}_{t+1}^{\text{fus}}$ via Eq. (14)
15: **end for**
16: **return** $\boldsymbol{\theta}_T$

---

not participated for many rounds, stale statistics are gradually regressed toward population averages. New clients are initialized with population-mean statistics and high staleness, which encourages early exploration without causing abrupt changes in the global latent state. Note that $q_i^{(t)}$ denotes selection probability, distinct from the sample weight $p_i = n_i / \sum_j n_j$ used in aggregation.

### 3.5. Trust-Weighted Aggregation

The final component addresses aggregation from heterogeneous clients, where prediction uncertainty $S_t$ serves as a calibration signal. We use modality-specific routing for encoders and uncertainty-calibrated weighting for the fusion layer. We do not claim modality-specific routing itself as a new aggregation principle; it is a necessary safeguard for heterogeneous multimodal FL, since clients lacking a modality should not update the corresponding encoder. The new role of FedSSM is to use the same surprise signal from the predictive model to calibrate fusion-layer trust.

**Modality-Specific Routing.** For encoder parameters, we aggregate only from clients possessing the corresponding modality:

$$\boldsymbol{\theta}_{t+1}^m = \frac{\sum_{i:m\in\mathcal{M}_i} n_i \boldsymbol{\theta}_{i,t}^m}{\sum_{j:m\in\mathcal{M}_j} n_j}, \quad \forall m \in \mathcal{M}. \quad (12)$$

This avoids injecting meaningless modality parameters from clients that do not own modality $m$. It also makes the aggregation compatible with both full-modality and partial-modality clients.

**Uncertainty-Calibrated Fusion.** For fusion layer parameters, we introduce trust weights calibrated by prediction uncertainty:

$$\tau_i = \exp\left(-S_t \cdot \left(\mathbb{1}_{|\mathcal{M}_i|<M} + \eta \frac{|L_i - \bar{L}_t|}{\text{std}(\{L_j\})}\right)\right), \quad (13)$$

where the first term reflects modality incompleteness and the second term penalizes outlier losses. This weight is surprise-gated. When $S_t$ is low, the training trajectory is predictable and $\tau_i \approx 1$, so partial-modality clients are not unnecessarily suppressed. When $S_t$ is high, the server becomes more cautious and down-weights updates that are more likely to destabilize fusion. Therefore, the method does not assume that every partial-modality client is low quality; it only reduces trust when the global dynamics are uncertain.

The fusion layer is updated via:

$$\boldsymbol{\theta}_{t+1}^{\text{fus}} = \frac{\sum_{i\in\mathcal{S}_t} n_i \tau_i \boldsymbol{\theta}_{i,t}^{\text{fus}}}{\sum_{i\in\mathcal{S}_t} n_i \tau_i}. \quad (14)$$

Algorithm 1 summarizes the complete FedSSM procedure. Theoretical convergence analysis is provided in Appendix A.

## 4. Experiments

### 4.1. Experimental Setup

**Datasets.** We evaluate FedSSM on four multimodal benchmarks (See Table 1 for details): (1) **VQA v2.0** (Goyal et al., 2017) requires joint reasoning over images and natural language questions for visual question answering; (2) **UCF-101** (Soomro et al., 2012) combines video frames with audio streams for human action recognition across 101 categories; (3) **Hateful Memes** (Kiela et al., 2020) pairs meme images with embedded text for multimodal hate speech detection; (4) **MiT-51** (Monfort et al., 2019), a subset of Moments in Time, integrates video and audio for fine-grained action classification.

*Table 1.* Summary of multimodal benchmark datasets.

| Dataset | Modalities | Samples | Classes |
|---|---|---|---|
| VQA v2.0 | I + T | 658K | 3,129 |
| UCF-101 | V + A | 13K | 101 |
| Hateful Memes | I + T | 10K | 2 |
| MiT-51 | V + A | 50K | 51 |

**Heterogeneity Simulation.** We partition data across $K = 100$ clients using Dirichlet distribution with concentration parameter $\alpha \in \{0.1, 0.5\}$, where smaller $\alpha$ indicates higher distribution heterogeneity. For modality heterogeneity, we designate 50% of clients as partial-modality clients, each

*Table 2.* Performance comparison across four multimodal benchmarks. All experiments use 100 clients and 50% partial-modality ratio. Baseline methods use a fixed 10% client participation per round, while FedSSM employs adaptive participation with $N_{\min} = 5$ and $N_{\max} = 15$ (average participation: 9.2% on VQA v2.0, 8.7% on UCF-101, 10.4% on Hateful Memes, 9.8% on MiT-51 under $\alpha = 0.1$). Results averaged over 3 runs. **Bold**: best; Underline: second best. [†]AUC-ROC (%) is reported for Hateful Memes; Accuracy (%) for others.

| Method | VQA v2.0 Image + Text | | UCF-101 Video + Audio | | Hateful Memes[†] Image + Text | | MiT-51 Video + Audio | |
|---|---|---|---|---|---|---|---|---|
| | $\alpha = 0.1$ | $\alpha = 0.5$ | $\alpha = 0.1$ | $\alpha = 0.5$ | $\alpha = 0.1$ | $\alpha = 0.5$ | $\alpha = 0.1$ | $\alpha = 0.5$ |
| FedAvg (McMahan et al., 2017) | $58.42_{\pm 0.72}$ | $64.17_{\pm 0.53}$ | $72.36_{\pm 0.61}$ | $76.84_{\pm 0.47}$ | $52.64_{\pm 0.85}$ | $57.38_{\pm 0.68}$ | $31.25_{\pm 0.79}$ | $34.82_{\pm 0.64}$ |
| Oort (Lai et al., 2021) | $61.35_{\pm 0.63}$ | $66.48_{\pm 0.46}$ | $75.18_{\pm 0.54}$ | $79.23_{\pm 0.41}$ | $55.82_{\pm 0.76}$ | $60.15_{\pm 0.61}$ | $33.87_{\pm 0.71}$ | $37.24_{\pm 0.56}$ |
| CriticalFL (Yan et al., 2023) | $63.72_{\pm 0.55}$ | $68.53_{\pm 0.40}$ | $77.45_{\pm 0.48}$ | $81.12_{\pm 0.36}$ | $58.47_{\pm 0.68}$ | $62.83_{\pm 0.54}$ | $35.94_{\pm 0.63}$ | $39.17_{\pm 0.49}$ |
| FedMSplit (Chen & Zhang, 2022) | $61.86_{\pm 0.61}$ | $67.04_{\pm 0.45}$ | $75.62_{\pm 0.52}$ | $79.75_{\pm 0.40}$ | $56.23_{\pm 0.74}$ | $60.74_{\pm 0.59}$ | $34.28_{\pm 0.69}$ | $37.65_{\pm 0.54}$ |
| CreamFL (Yu et al., 2023) | $63.45_{\pm 0.56}$ | $68.27_{\pm 0.41}$ | $77.14_{\pm 0.49}$ | $80.86_{\pm 0.37}$ | $58.15_{\pm 0.69}$ | $62.46_{\pm 0.55}$ | $35.62_{\pm 0.64}$ | $38.85_{\pm 0.50}$ |
| mmFedMC (Yuan et al., 2024) | $64.18_{\pm 0.53}$ | $68.94_{\pm 0.39}$ | $78.27_{\pm 0.46}$ | $81.73_{\pm 0.35}$ | $59.38_{\pm 0.65}$ | $63.72_{\pm 0.52}$ | $36.75_{\pm 0.60}$ | $39.94_{\pm 0.47}$ |
| FedMBridge (Chen & Zhang, 2024) | $65.23_{\pm 0.50}$ | $69.65_{\pm 0.37}$ | $79.58_{\pm 0.43}$ | $82.94_{\pm 0.33}$ | $61.24_{\pm 0.61}$ | $65.18_{\pm 0.49}$ | $38.13_{\pm 0.56}$ | $41.26_{\pm 0.44}$ |
| **FedSSM (Ours)** | **$68.47_{\pm 0.44}$** | **$72.13_{\pm 0.32}$** | **$82.85_{\pm 0.37}$** | **$85.67_{\pm 0.28}$** | **$65.72_{\pm 0.53}$** | **$68.93_{\pm 0.43}$** | **$41.86_{\pm 0.49}$** | **$44.53_{\pm 0.38}$** |
| $\Delta$ *vs.* FedMBridge | +3.24 | +2.48 | +3.27 | +2.73 | +4.48 | +3.75 | +3.73 | +3.27 |

possessing a uniformly sampled subset of available modalities with at least one modality present. Quantity heterogeneity is introduced by sampling client data sizes from a log-normal distribution ($\mu = 0$, $\sigma = 1$), then scaling to allocate the total dataset, yielding size ratios from approximately $0.1\times$ to $5\times$ the average.

**Baselines.** We compare against three families of methods: (1) *Random selection*: FedAvg (McMahan et al., 2017); (2) *Unimodal client selection*: Oort (Lai et al., 2021) and CriticalFL (Yan et al., 2023), representing utility-based and learning-based selection respectively; (3) *Multimodal federated learning*: FedMSplit (Chen & Zhang, 2022), CreamFL (Yu et al., 2023), mmFedMC (Yuan et al., 2024), and FedMBridge (Chen & Zhang, 2024). We focus on methods addressing multimodal heterogeneity or demonstrating strong performance in heterogeneous settings. Additional comparisons with Power-of-Choice (Cho et al., 2022), FedProx (Li et al., 2020), and FAVOR (Wang et al., 2020) are in Appendix B.

**Baseline Configuration.** For unimodal selection methods (Oort, CriticalFL), we combine their selection mechanisms with modality-specific encoder aggregation (Eq. (12)) to handle modality heterogeneity. All baseline hyperparameters are tuned via grid search on a held-out validation set (10% of training data) using VQA v2.0 with $\alpha = 0.3$, then fixed across experiments. For Oort, we search over utility threshold $\in \{0.5, 0.7, 0.9\}$ and exploration factor $\in \{0.1, 0.2, 0.3\}$. For CriticalFL, we search over critical client ratio $\in \{0.1, 0.2, 0.3\}$. Multimodal baselines use configurations from their original papers.

**Implementation Details.** For image-text tasks (VQA v2.0, Hateful Memes), we use ResNet-18 as the image encoder and a single-layer LSTM (256 hidden units) as the text encoder; for video-audio tasks (UCF-101, MiT-51), we

use ResNet-18 with temporal average pooling over 16 sampled frames and a single-layer LSTM on log-mel spectrograms. All tasks employ a two-layer MLP (512-256) for fusion, trained with SGD (lr=0.01, momentum=0.9, weight decay=$10^{-4}$, batch size=32) for 5 local epochs per round over 100 rounds. FedSSM parameters are: $N_{\min} = 5$, $N_{\max} = 15$, $\kappa = 2.0$, $\beta_1 = 0.5$, $\beta_2 = 0.3$, $\beta_3 = 1.0$, $\tau = 5.0$, $\eta = 0.5$, SSM dimension $d = 64$, smoothing $\beta = 0.9$, forgetting $\lambda = 0.95$, staleness decay $\gamma = 0.9$, with SSM updated via Adam (lr=$10^{-3}$). Results are averaged over 3 runs.

### 4.2. Main Results

**Performance Comparison.** Table 2 presents results under varying heterogeneity. FedSSM consistently achieves the best performance, outperforming the strongest baseline FedMBridge by 2.48%–4.48%. Conventional strategies like Oort (Lai et al., 2021) and CriticalFL (Yan et al., 2023) improve upon FedAvg by 3–6% via utility-based heuristics, yet remain limited by model-free selection that ignores how current decisions shape future dynamics. Multimodal methods such as FedMSplit (Chen & Zhang, 2022), CreamFL (Yu et al., 2023), and mmFedMC (Yuan et al., 2024) mitigate heterogeneity through adaptive splitting or compensation, while FedMBridge (Chen & Zhang, 2024) establishes a strong baseline via topology-aware hypernetworks. The performance gap becomes particularly pronounced under severe heterogeneity ($\alpha = 0.1$), where FedSSM surpasses FedMBridge by 4.48% on Hateful Memes and 3.73% on MiT-51. High distribution heterogeneity induces volatile optimization dynamics with large gradient variance across rounds, causing model-free methods to lag behind sudden shifts. FedSSM addresses this by predicting future states through the decision-aware SSM, enabling proactive adaptation before performance degradation occurs. On UCF-

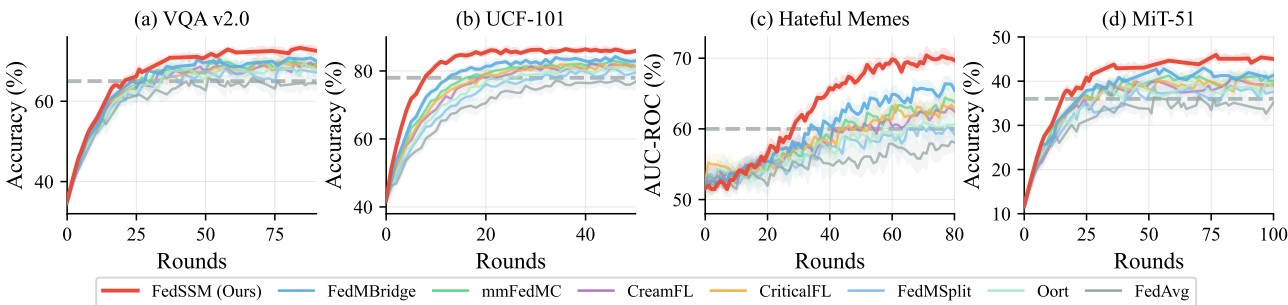

*Figure 3.* Convergence curves on four multimodal benchmarks under $\alpha = 0.1$. Dashed lines indicate target accuracy thresholds. FedSSM consistently reaches target accuracy with fewer communication rounds.

*Table 3.* Communication efficiency analysis on VQA v2.0 under high heterogeneity ($\alpha = 0.1$). Cost (GB) measures total communication to reach 65% accuracy. Time/R (s) denotes wall-clock time per round on a server with NVIDIA A100 GPU.

| Metric | FedAvg | Oort | CriticalFL | FedMSplit | CreamFL | mmFedMC | FedMBridge | **FedSSM** |
|---|---|---|---|---|---|---|---|---|
| Cost (GB)↓ | 3.50 | 3.07 | 2.64 | 3.26 | 2.83 | 2.45 | 2.16 | **1.14** |
| Time/R (s)↓ | 51.8 | 54.7 | 56.3 | 61.4 | 65.2 | 67.8 | 70.5 | **45.2** |

*Table 4.* Ablation study of FedSSM components on VQA v2.0 and UCF-101 ($\alpha = 0.1$). We report accuracy (%) and rounds to reach target.

| Variant | VQA v2.0 | | UCF-101 | |
|---|---|---|---|---|
| | Acc (%) | Rounds | Acc (%) | Rounds |
| **FedSSM (Full)** | **68.47**±0.38 | **22**±2 | **82.85**±0.35 | **14**±1 |
| w/o Decision-Aware SSM | 65.18±0.52 | 38±3 | 79.24±0.48 | 26±2 |
| w/o Counterfactual Selection | 66.32±0.45 | 32±2 | 80.41±0.42 | 22±2 |
| w/o Adaptive Budget | 67.15±0.41 | 28±2 | 81.28±0.38 | 18±1 |
| w/o Trust-Weighted Fusion | 67.62±0.40 | 25±2 | 81.73±0.37 | 16±1 |
| Random Selection | 63.58±0.58 | 48±4 | 77.46±0.55 | 34±3 |

101, FedSSM achieves consistent improvements of 3.27% ($\alpha = 0.1$) and 2.73% ($\alpha = 0.5$) over FedMBridge, demonstrating effective capture of complementary video-audio information. On MiT-51, where centralized training with the same architecture achieves 38.2% and the original paper (Monfort et al., 2019) reports 35.9%, FedSSM attains 41.86% under high heterogeneity, approaching centralized performance despite severe data fragmentation.

**Convergence Analysis.** Figure 3 illustrates convergence trajectories under high heterogeneity ($\alpha = 0.1$). FedSSM reaches higher final accuracy with substantially fewer rounds. On UCF-101, FedSSM achieves 78% accuracy within 14 rounds, while FedMBridge requires 21 rounds and Oort requires 31 rounds. This acceleration stems from the adaptive budget mechanism: when surprise $S_t$ is high during early training, FedSSM increases participation to acquire information rapidly, then reduces participation as training stabilizes. The consistent early-phase advantage across all benchmarks demonstrates that proactive inference enables efficient navigation of the optimization landscape.

**Communication Efficiency.** Table 3 reports total communication cost to reach target accuracy and time per round on VQA v2.0 under $\alpha = 0.1$. FedSSM achieves the lowest cost of 1.14 GB, reducing transmission by 47% versus FedMBridge and 67% versus FedAvg. Among model-free methods, CriticalFL and Oort reduce costs by identifying critical periods or high-utility clients, yet plateau around 2.6–3.1 GB as they cannot predict when reduced participation suffices. FedSSM's efficiency gain arises from two mechanisms: adaptive budgeting scales participation based on surprise, avoiding unnecessary communication when dynamics are predictable, while counterfactual selection identifies high-value clients maximizing information gain per round. FedSSM also achieves the fastest per-round time of 45.2 seconds despite additional SSM computations, because reduced participation more than compensates for lightweight state updates.

### 4.3. Analysis

**Scalability and Robustness.** Figure 4(a-b) examines scalability by varying the number of clients from 20 to 200. FedSSM maintains the highest accuracy and requires the fewest rounds across all scales, as counterfactual selection efficiently identifies informative clients even when the candidate pool expands. Figure 4(c-d) evaluates robustness by varying the partial-modality ratio from 0% to 75%. While FedAvg suffers accuracy degradation exceeding 20% at 75% ratio, FedSSM experiences only marginal decline of approximately 10%, benefiting from trust-weighted fusion that down-weights partial-modality clients when surprise is elevated.

**Ablation Study.** Table 4 investigates each component's contribution on VQA v2.0 and UCF-101 under $\alpha = 0.1$.

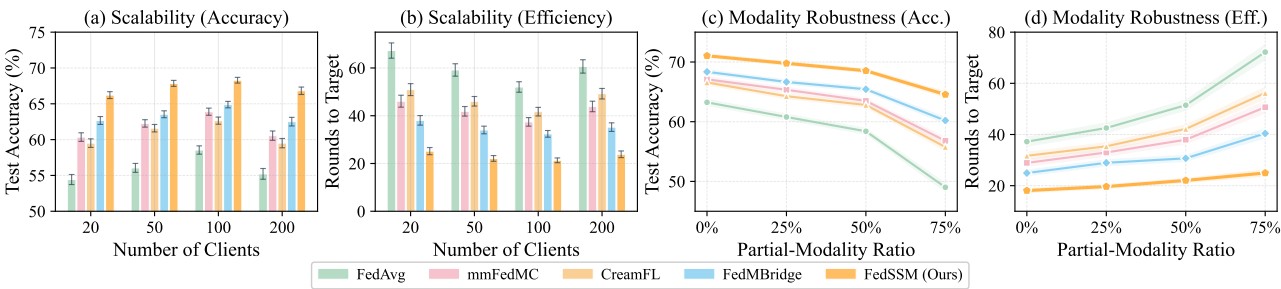

*Figure 4.* Scalability and robustness analysis on VQA v2.0 ($\alpha = 0.1$). (a-b) Effect of varying the number of clients from 20 to 200. (c-d) Effect of varying the partial-modality ratio from 0% to 75%.

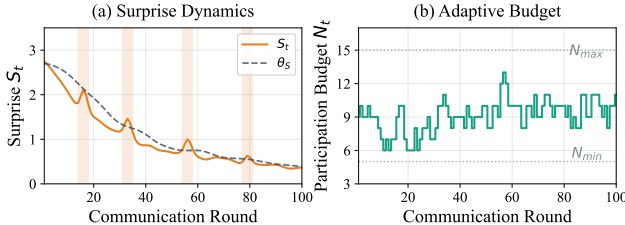

*Figure 5.* Proactive mechanism dynamics on VQA v2.0 ($\alpha = 0.1$). (a) Surprise $S_t$ and threshold $\theta_S$. (b) Adaptive participation budget $N_t$.

*Table 5.* Computational overhead analysis per round.

| Component | Time (s) | Memory (MB) |
|---|---|---|
| SSM State Update | 0.3 | 1.5 |
| Counterfactual Reasoning | 0.4 | 0.6 |
| Client Selection | 0.1 | 0.2 |
| **Total Overhead** | **0.8** | **2.3** |

Removing the decision-aware SSM causes the largest degradation, increasing rounds to target from 22 to 38 on VQA v2.0 and from 14 to 26 on UCF-101, confirming that modeling action-conditioned dynamics is essential. Counterfactual selection contributes the second-largest impact by enabling evaluation of alternative strategies; removing it increases convergence rounds by 45% on VQA v2.0 and 57% on UCF-101. Disabling adaptive budget results in moderate degradation (rounds increasing to 28 and 18), indicating dynamic participation accelerates learning under high uncertainty. Trust-weighted fusion shows smaller but consistent contributions, more pronounced on VQA v2.0 where cross-modal alignment is challenging. Random selection reduces performance to 63.58% and 77.46%, approaching model-free baselines. This variant uses uniform client sampling while retaining the same modality-specific routing and fusion design as FedSSM; therefore, its gap over vanilla FedAvg should be attributed to the aggregation backbone rather than to the selection policy.

**Visualization of Proactive Mechanism.** Figure 5 visualizes FedSSM's internal dynamics on VQA v2.0. In Figure 5(a), surprise $S_t$ starts at approximately 2.7 during early training when the SSM encounters unfamiliar terrain, then stabilizes below 0.5 after round 80. Shaded regions indicate transient spikes from sudden shifts in dynamics, triggering immediate budget responses. Figure 5(b) shows the adaptive behavior: $N_t$ fluctuates between 6 and 12 clients, elevating when $S_t$ exceeds $\theta_S$ to gather information and reducing when

predictions become accurate.

**Computational Overhead.** Table 5 reports FedSSM's overhead. The SSM update and counterfactual reasoning add approximately 0.8 seconds per round on a single CPU, negligible compared to local training time (30–60 seconds per client). Memory footprint requires 2.3 MB, independent of client population size. Compared to learning-based methods like FAVOR (Wang et al., 2020) requiring GPU-based policy networks, FedSSM achieves superior performance with lower requirements, making it practical for resource-constrained deployments. Extended experiments are in Appendix B.

## 5. Conclusion

We presented FedSSM, a framework reconceptualizing client selection as proactive inference rather than reactive response. The decision-aware state space model predicts training dynamics for counterfactual reasoning over candidate strategies. The surprise signal governs adaptive participation budgets based on uncertainty and calibrates trust weights in aggregation. Experiments on four benchmarks show FedSSM outperforms state-of-the-art by 2.5–4.5% while reducing communication rounds by over 30%. Limitations remain. The framework assumes stationary client populations; dynamic scenarios require online adaptation. Counterfactual reasoning scales linearly with action space size, limiting applicability to very large client pools. Future work could explore hierarchical state space models capturing both client-level and global dynamics.

## Acknowledgements

This work was supported by the National Natural Science Foundation of China under Grant No. 62501287 and No. 62401196, and the Natural Science Foundation of Jiangsu Province under Grant No. BK20240663.

## Impact Statement

This paper presents work whose goal is to advance the field of Machine Learning, specifically in optimizing client selection for multimodal federated learning systems. Our framework improves communication efficiency while preserving data privacy across distributed clients. There are many potential societal consequences of our work, none of which we feel must be specifically highlighted here.

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

# A. Theoretical Analysis of Proactive Convergence

We provide the convergence analysis of FedSSM under non-convex objectives and heterogeneous client distributions. The goal of this appendix is not to claim that the SSM changes the basic federated optimization rate, but to clarify why the proactive mechanism is theoretically meaningful. The analysis shows three points. First, the adaptive client sampling remains unbiased when importance weighting is used. Second, the surprise signal controls the unanticipated part of gradient heterogeneity and therefore justifies adaptive budgeting. Third, the online prediction error of the SSM and the action regret over the finite action space can be bounded, making the comparison with bandit-style selectors more explicit.

## A.1. Preliminaries and Assumptions

Let $\mathcal{F}(\boldsymbol{\theta}) = \sum_{i=1}^{K} p_i F_i(\boldsymbol{\theta})$ denote the global objective, where $p_i = n_i / \sum_j n_j$. At round $t$, the server samples a client set $\mathcal{S}_t$ according to the selection distribution induced by FedSSM. We denote by $q_i^{(t)}$ the sampling probability used for client $i$ at round $t$. The stochastic gradient returned by client $i$ is denoted as $\nabla F_i(\boldsymbol{\theta}_t; \xi_i)$.

**Assumption A.1** ($L$-Smoothness). Each local objective $F_i$ is $L$-smooth, i.e., $\|\nabla F_i(\boldsymbol{\theta}) - \nabla F_i(\boldsymbol{\phi})\| \leq L\|\boldsymbol{\theta} - \boldsymbol{\phi}\|$ for all $\boldsymbol{\theta}$ and $\boldsymbol{\phi}$.

**Assumption A.2** (Bounded Local Variance). For each client $i$, the stochastic gradient satisfies $\mathbb{E}_\xi[\nabla F_i(\boldsymbol{\theta}; \xi)] = \nabla F_i(\boldsymbol{\theta})$ and $\mathbb{E}_\xi[\|\nabla F_i(\boldsymbol{\theta}; \xi) - \nabla F_i(\boldsymbol{\theta})\|^2] \leq \sigma_l^2$.

**Assumption A.3** (Bounded Client Heterogeneity). There exists $\zeta > 0$ such that $\sum_{i=1}^{K} p_i \|\nabla F_i(\boldsymbol{\theta}) - \nabla \mathcal{F}(\boldsymbol{\theta})\|^2 \leq \zeta^2$ for all $\boldsymbol{\theta}$.

**Assumption A.4** (Minimum Exploration). There exists $q_{\min} > 0$ such that $q_i^{(t)} \geq q_{\min}$ for all clients and all rounds.

Assumption A.4 is standard for adaptive client selection. In FedSSM, it is enforced by the exploration term in Eq. (11). This avoids degeneracy caused by assigning zero probability to stale or rarely selected clients.

## A.2. Unbiased Aggregation under Adaptive Selection

The selection distribution of FedSSM changes across rounds because it depends on the surprise signal, staleness, loss statistics, and the exploration-exploitation mixture. We therefore analyze the importance-weighted gradient estimator:

$$\bar{\mathbf{g}}_t = \frac{1}{N_t} \sum_{i \in \mathcal{S}_t} \frac{p_i}{q_i^{(t)}} \nabla F_i(\boldsymbol{\theta}_t; \xi_i). \tag{15}$$

**Lemma A.5** (Unbiased Gradient Estimator). *Under adaptive client selection, the estimator in Eq. (15) satisfies* $\mathbb{E}[\bar{\mathbf{g}}_t \mid \boldsymbol{\theta}_t] = \nabla \mathcal{F}(\boldsymbol{\theta}_t)$.

*Proof.* Taking expectation over both client sampling and local stochasticity gives

$$\mathbb{E}[\bar{\mathbf{g}}_t \mid \boldsymbol{\theta}_t] = \sum_{i=1}^{K} q_i^{(t)} \frac{p_i}{q_i^{(t)}} \nabla F_i(\boldsymbol{\theta}_t). \tag{16}$$

The right hand side equals $\sum_{i=1}^{K} p_i \nabla F_i(\boldsymbol{\theta}_t)$, which is exactly $\nabla \mathcal{F}(\boldsymbol{\theta}_t)$. $\square$

This lemma shows that FedSSM changes the sampling distribution but not the target objective. The adaptive policy affects the variance of the estimator, not its expectation.

## A.3. Variance Decomposition

The variance of Eq. (15) contains two terms: stochastic local gradient noise and heterogeneity-induced sampling noise.

**Lemma A.6** (Variance Bound). *Under Assumptions A.2 and A.4, the gradient estimator satisfies*

$$\mathbb{E}[\|\bar{\mathbf{g}}_t - \nabla \mathcal{F}(\boldsymbol{\theta}_t)\|^2] \leq \frac{\sigma_l^2}{N_t} \sum_{i=1}^{K} \frac{p_i^2}{q_i^{(t)}} + \frac{1}{N_t} \sum_{i=1}^{K} \frac{p_i^2}{q_i^{(t)}} \|\nabla F_i(\boldsymbol{\theta}_t) - \nabla \mathcal{F}(\boldsymbol{\theta}_t)\|^2. \tag{17}$$

*Proof.* Using Lemma A.5, the variance can be decomposed into local stochastic variance and client sampling variance. The local term is bounded by Assumption A.2. The sampling term follows from the importance-weighted form in Eq. (15). Combining both terms gives Eq. (17). □

The first term decreases with the participation budget $N_t$. The second term is more important in heterogeneous MMFL because it depends on the disagreement between local and global gradients. FedSSM aims to reduce this term by increasing participation when the predicted training trajectory becomes unreliable.

### A.4. Surprise as Unanticipated Heterogeneity

FedSSM does not assume that the total heterogeneity is directly observable before client selection. Instead, it estimates whether the observed heterogeneity was anticipated by the SSM. Let the server observation be $\mathbf{x}_t = [L_t, \|\nabla L_t\|, \sigma_t^2, t/T]^\top$, where $\sigma_t^2$ is the observed inter-client gradient variance in the selected set. Let $\hat{\mathbf{x}}_t$ be the SSM prediction, and let $\hat{\sigma}_t^2$ denote its third component.

**Proposition A.7** (Surprise Controls Unanticipated Variance). *Let $S_t = (\mathbf{x}_t - \hat{\mathbf{x}}_t)^\top \boldsymbol{\Sigma}^{-1} (\mathbf{x}_t - \hat{\mathbf{x}}_t)$ be the surprise score. Then the unanticipated variance satisfies*

$$|\sigma_t^2 - \hat{\sigma}_t^2| \leq \sqrt{\Sigma_{33}} \sqrt{S_t}, \tag{18}$$

*where $\Sigma_{33}$ is the third diagonal entry of $\boldsymbol{\Sigma}$.*

*Proof.* By the definition of the Mahalanobis distance, the contribution of the third component is upper bounded by the full surprise score. Therefore,

$$S_t \geq \frac{(\sigma_t^2 - \hat{\sigma}_t^2)^2}{\Sigma_{33}}. \tag{19}$$

Taking square roots yields Eq. (18). □

This proposition gives the main theoretical meaning of surprise. A large $S_t$ does not simply mean that the system is heterogeneous. It means that the current heterogeneity is not well predicted by the learned dynamics. This distinction matters because high but predictable heterogeneity can be handled by the existing selection policy, while unanticipated heterogeneity requires additional exploration or participation.

### A.5. Adaptive Budget and Variance Compensation

FedSSM adapts the participation budget according to the previous surprise score:

$$N_t = N_{\min} + \left\lfloor \frac{N_{\max} - N_{\min}}{1 + \exp(-\kappa(S_{t-1} - \theta_S))} \right\rfloor. \tag{20}$$

The role of Eq. (20) is to allocate more clients when the previous round reveals unanticipated dynamics. This does not require the server to know the exact client heterogeneity in advance.

**Lemma A.8** (Variance Compensation by Adaptive Budgeting). *Under Assumptions A.2–A.4, there exists a constant $C_{\mathrm{var}} > 0$ such that*

$$\mathbb{E}[\|\bar{\mathbf{g}}_t - \nabla\mathcal{F}(\boldsymbol{\theta}_t)\|^2] \leq \frac{C_{\mathrm{var}}}{N_t}. \tag{21}$$

*Moreover, $N_t$ is non-decreasing in $S_{t-1}$, so the variance factor $1/N_t$ decreases when the previous surprise increases.*

*Proof.* By Eq. (17), Assumption A.4, and Assumption A.3, both variance components are bounded by constants independent of $t$. Specifically, $\sum_i p_i^2/q_i^{(t)} \leq \|p\|_2^2/q_{\min}$ and the heterogeneity term is bounded by $\zeta^2/q_{\min}$. Hence Eq. (21) holds for a constant $C_{\mathrm{var}}$ depending on $\sigma_l^2$, $\zeta^2$, $p$, and $q_{\min}$. The monotonicity follows directly from the sigmoid function in Eq. (20). □

Lemma A.8 gives a conservative but useful statement: FedSSM preserves bounded variance under adaptive sampling and reduces the variance scale when surprise increases. Proposition A.7 further explains why surprise is the right signal to trigger this increase.

## A.6. Online Prediction Error of the SSM

A concern with the original convergence bound is the presence of the cumulative SSM error term. We make this term explicit through a standard online prediction argument. Let $\phi_t$ denote the SSM parameters at round $t$, and define the prediction loss:

$$\ell_t(\phi_t) = \|\mathbf{x}_t - \hat{\mathbf{x}}_t(\phi_t)\|^2. \tag{22}$$

**Assumption A.9** (Bounded Online Prediction). The SSM parameter domain has diameter $D$, and the gradient of $\ell_t(\phi)$ is bounded by $G$ for all $t$. The observation vector is normalized such that $\|\mathbf{x}_t\| \leq B$.

FedSSM updates the SSM online with exponential forgetting. Let $H = 1/(1 - \lambda)$ be the effective memory horizon.

**Lemma A.10** (Cumulative SSM Prediction Error). *Under Assumption A.9, online gradient updates with forgetting factor $\lambda$ satisfy*

$$\sum_{t=1}^{T} \ell_t(\phi_t) \leq \min_{\phi} \sum_{t=1}^{T} \ell_t(\phi) + O(DG\sqrt{TH}). \tag{23}$$

*Proof.* This follows from the standard regret bound of online gradient descent with exponentially weighted losses (Hazan, 2016). The forgetting factor restricts the effective horizon to $H = 1/(1 - \lambda)$, giving the stated dependence on $T$ and $H$.    □

Let $\epsilon_* = \frac{1}{T} \min_{\phi} \sum_{t=1}^{T} \ell_t(\phi)$ be the average loss of the best fixed SSM predictor in hindsight. Lemma A.10 implies

$$\frac{1}{T} \sum_{t=1}^{T} \ell_t(\phi_t) \leq \epsilon_* + O(DG\sqrt{H/T}). \tag{24}$$

This makes the SSM approximation term explicit. When the training dynamics are predictable by the chosen SSM class, $\epsilon_*$ is small; otherwise, the final convergence bound degrades gracefully with the average prediction error.

## A.7. Main Convergence Result

**Theorem A.11** (Convergence of FedSSM). *Under Assumptions A.1–A.9, let the learning rate satisfy $\eta \leq 1/(4L)$. Then FedSSM satisfies*

$$\frac{1}{T} \sum_{t=0}^{T-1} \mathbb{E}[\|\nabla\mathcal{F}(\boldsymbol{\theta}_t)\|^2] \leq \frac{4(\mathcal{F}(\boldsymbol{\theta}_0) - \mathcal{F}^*)}{\eta T} + 2L\eta C_{\text{var}} + 2L\eta C_{\text{ssm}} \left( \epsilon_* + O(DG\sqrt{H/T}) \right), \tag{25}$$

*where $C_{\text{var}}$ is from Lemma A.8, and $C_{\text{ssm}}$ is a constant that maps SSM prediction error to the residual unanticipated variance.*

*Proof.* By $L$-smoothness, we have

$$\mathcal{F}(\boldsymbol{\theta}_{t+1}) \leq \mathcal{F}(\boldsymbol{\theta}_t) - \eta\langle\nabla\mathcal{F}(\boldsymbol{\theta}_t), \bar{\mathbf{g}}_t\rangle + \frac{L\eta^2}{2}\|\bar{\mathbf{g}}_t\|^2. \tag{26}$$

Taking expectation and using Lemma A.5 gives a descent inequality with a variance penalty. Lemma A.8 bounds the sampling and stochastic variance by $C_{\text{var}}/N_t$, which is absorbed into $C_{\text{var}}$. The remaining residual comes from the mismatch between the predicted and observed training state. By Proposition A.7, this mismatch is controlled by the SSM prediction error. Lemma A.10 then gives the average prediction error bound in Eq. (24). Summing the descent inequality over $t = 0, \ldots, T - 1$ and rearranging yields Eq. (25).    □

**Corollary A.12** (Rate under Bounded Prediction Error). *With $\eta = O(1/\sqrt{T})$, FedSSM achieves*

$$\frac{1}{T} \sum_{t=0}^{T-1} \mathbb{E}[\|\nabla\mathcal{F}(\boldsymbol{\theta}_t)\|^2] = O(1/\sqrt{T}) + O(\epsilon_*/\sqrt{T}). \tag{27}$$

*When the SSM class tracks the training dynamics well, i.e., $\epsilon_*$ is small, the standard non-convex FL rate $O(1/\sqrt{T})$ is preserved.*

This result clarifies the role of the SSM error term. FedSSM does not require perfect prediction. It only requires that the online predictor maintains a bounded average error, and the additional term decreases with the FL learning rate.

## A.8. Regret over the Finite Action Space

We also provide a regret view to make the comparison with bandit-style selection more direct. Let $\mathcal{A}$ be the finite action space used by FedSSM, where each action specifies a participation budget and an exploration-exploitation level. Let $u_t(\mathbf{a})$ be the true utility of action $\mathbf{a}$ at round $t$, and let $\hat{u}_t(\mathbf{a})$ be the SSM-predicted utility used by FedSSM.

**Assumption A.13** (Lipschitz Utility). The utility function is $G_u$-Lipschitz with respect to the predicted observation, i.e., $|u_t(\mathbf{a}) - \hat{u}_t(\mathbf{a})| \leq G_u \|\mathbf{x}_{t+1}^{(\mathbf{a})} - \hat{\mathbf{x}}_{t+1}^{(\mathbf{a})}\|$ for all $\mathbf{a} \in \mathcal{A}$.

Let $\mathbf{a}_t$ be the action selected by FedSSM and $\mathbf{a}_t^* = \arg\max_{\mathbf{a} \in \mathcal{A}} u_t(\mathbf{a})$ be the one-step oracle action.

**Proposition A.14** (Model-Based Action Regret). *Under Assumption A.13, the dynamic regret of FedSSM satisfies*

$$R_T = \sum_{t=1}^{T} [u_t(\mathbf{a}_t^*) - u_t(\mathbf{a}_t)] \leq 2G_u \sum_{t=1}^{T} \max_{\mathbf{a} \in \mathcal{A}} \|\mathbf{x}_{t+1}^{(\mathbf{a})} - \hat{\mathbf{x}}_{t+1}^{(\mathbf{a})}\|. \tag{28}$$

*Proof.* Since FedSSM selects the action maximizing the predicted utility, $\hat{u}_t(\mathbf{a}_t) \geq \hat{u}_t(\mathbf{a}_t^*)$. Therefore,

$$u_t(\mathbf{a}_t^*) - u_t(\mathbf{a}_t) \leq |u_t(\mathbf{a}_t^*) - \hat{u}_t(\mathbf{a}_t^*)| + |\hat{u}_t(\mathbf{a}_t) - u_t(\mathbf{a}_t)|. \tag{29}$$

Applying Assumption A.13 to both terms gives the result after summing over $t$. □

Proposition A.14 highlights the distinction between FedSSM and a standard multi-armed bandit. A bandit method reduces regret by repeatedly observing realized rewards for selected actions. FedSSM reduces regret by improving a predictive model of the training dynamics, so its regret is controlled by the counterfactual prediction error. Combining Proposition A.14 with Lemma A.10 yields sublinear model-based regret when the SSM prediction error is sublinear over the effective online horizon.

## A.9. Comparison with Reactive Selection

Reactive selection methods, such as utility-based or bandit-style client sampling, typically update action scores after observing realized feedback. Their variance reduction depends on how quickly past rewards reveal the current client utility. In rapidly changing MMFL, this can be delayed because modality coverage, label skew, and quantity imbalance jointly affect the next training state.

FedSSM uses a different inductive bias. It models the transition from the current latent state to the next observation under candidate actions, and then uses the prediction error as a shared signal for budgeting, exploration, and aggregation trust. Theoretical results above show that this design preserves unbiased aggregation, compensates variance through adaptive participation, and admits explicit prediction-error and action-regret bounds.

**Remark.** The analysis focuses on one local update step for clarity. Multiple local steps can be handled by adding the standard client-drift term used in FedAvg-type analyses (Li et al., 2020; Karimireddy et al., 2020). This extension does not change the role of the surprise signal; it only adds a drift residual to the variance term.

# B. More Experiments

This section provides additional experiments and analysis to further examine FedSSM from three perspectives. First, we clarify the empirical distinction between FedSSM and bandit-style selection. Second, we expand the hyperparameter sensitivity study to all major hyperparameters. Third, we report additional results under fair participation, dataset-level heterogeneity, and three-modality settings.

## B.1. Comparison with Bandit-Style Client Selection

FedSSM is related to online decision making, but it is different from standard multi-armed bandit or contextual bandit client selection. In a bandit formulation, each action $a \in \mathcal{A}$ is treated as an arm, and the learner updates its action value after observing the realized reward:

$$Q_{t+1}(a_t) = Q_t(a_t) + \alpha_t(r_t - Q_t(a_t)). \tag{30}$$

*Table 6.* Conceptual comparison between bandit-style selection and FedSSM.

| Aspect | Bandit / UCB | Contextual bandit | FedSSM |
|---|---|---|---|
| Object being learned | Arm reward | Context-action reward | State transition |
| Uses current state/context | Limited | Yes | Yes |
| Predicts next optimization state | No | No | Yes |
| Uncertainty source | Arm confidence | Reward uncertainty | Trajectory surprise |
| Counterfactual action evaluation | Reward-level | Reward-level | State-level |
| Controls participation budget | Extra rule | Extra rule | Native |
| Controls aggregation trust | Extra rule | Extra rule | Native |
| Main feedback signal | Realized reward | Realized reward | Prediction error and reward |

*Table 7.* Comparison with representative reactive and learning-based selection baselines under $\alpha = 0.1$. Hateful Memes reports AUC-ROC, and other datasets report accuracy. "Gain" denotes the absolute improvement of FedSSM over the strongest baseline in each row.

| Dataset | Metric | Power-of-Choice | FedProx | FAVOR | FedSSM | Gain |
|---|---|---|---|---|---|---|
| VQA v2.0 | Acc. | 62.85±0.54 | 59.73±0.62 | 64.27±0.47 | **68.47±0.44** | +4.20 |
| UCF-101 | Acc. | 78.13±0.51 | 76.21±0.58 | 80.95±0.46 | **82.85±0.37** | +1.90 |
| H. Memes | AUC | 59.62±0.69 | 57.84±0.73 | 62.84±0.61 | **65.72±0.53** | +2.88 |
| MiT-51 | Acc. | 37.42±0.68 | 35.68±0.75 | 40.18±0.63 | **41.86±0.49** | +1.68 |

For example, $r_t$ can be defined as the validation improvement, loss reduction, or communication-adjusted utility after selecting a client set. This formulation is useful for online exploration, but it mainly learns which action has produced high reward in the past. The transition of the training state itself remains implicit.

Contextual bandits further condition the action value on the current context $\mathbf{x}_t$, often written as $Q(\mathbf{x}_t, a)$. However, the objective is still to estimate the immediate reward of an action under the current context. It does not explicitly answer the following question: if we choose this budget and this client mixture now, what will the next optimization state look like? This distinction is important in multimodal FL because the reward is highly non-stationary. The same client can be useful in an early round, redundant after its modality has been sufficiently learned, or harmful when its local distribution amplifies fusion instability.

FedSSM uses a different inductive bias. It models the transition of the training trajectory under candidate actions:

$$\tilde{\mathbf{x}}_{t+1}^{(\mathbf{a})} = \mathbf{C} \left[ \bar{\mathbf{A}}(\mathbf{a})\mathbf{h}_t + \bar{\mathbf{B}}(\mathbf{a})\hat{\mathbf{x}}_t \right]. \tag{31}$$

Thus, FedSSM does not only update an arm value after observing a reward. It first predicts the next optimization state under candidate actions and then selects the action with the most favorable predicted consequence. This makes the method closer to a lightweight model-based controller than to a pure bandit selector.

The uncertainty signals are also different. In UCB-style bandits, uncertainty is usually arm-level statistical uncertainty, such as a confidence bonus proportional to $\sqrt{\log t / n_t(a)}$. In FedSSM, uncertainty is trajectory-level prediction error, measured by the surprise score $S_t$. This surprise signal reflects how much the actual training state deviates from the predicted state. As a result, the same signal can jointly control three decisions: the participation budget $N_t$, the exploration-exploitation mixture $\rho_t$, and the aggregation trust weight $\tau_i$. A bandit-style method can be extended with similar heuristics, but these controls are usually designed separately through reward shaping or additional rules.

Table 6 summarizes the conceptual difference. The key point is not that bandit methods are unsuitable, but that FedSSM models a different object: it predicts state transitions rather than only estimating action rewards.

The performance gap is especially clear on VQA v2.0 and Hateful Memes, where modality and label imbalance are strongly coupled. This supports our claim that the advantage of FedSSM does not merely come from adaptive exploration, but from using a predictive state model to anticipate how selection decisions affect the next training state.

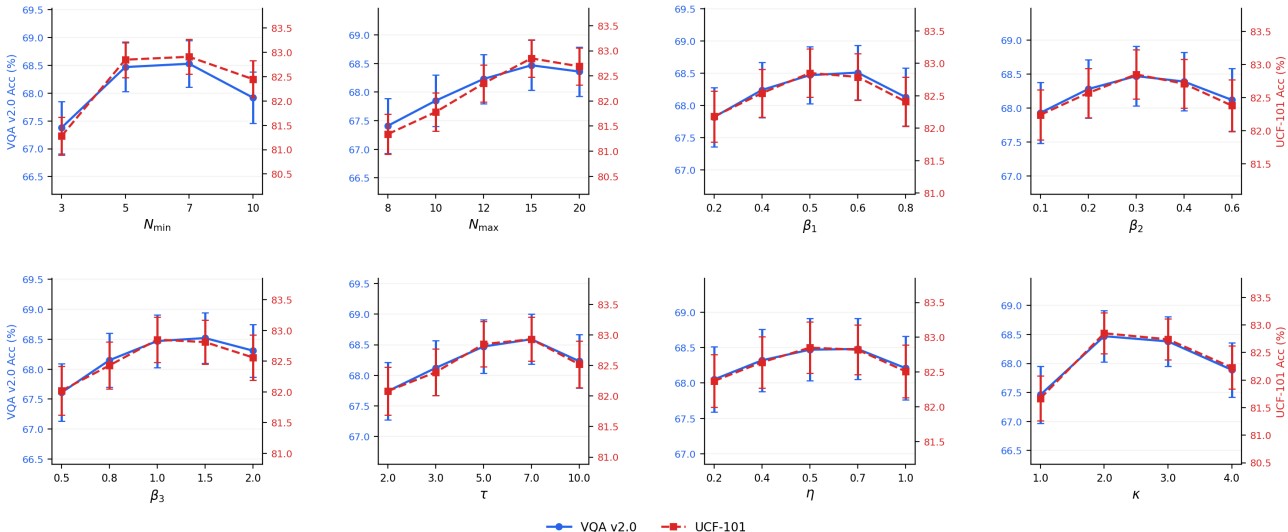

*Figure 6.* Hyperparameter sensitivity of FedSSM on VQA v2.0 and UCF-101. Blue curves correspond to VQA v2.0 and red curves correspond to UCF-101. Error bars denote $\pm 1$ standard deviation over 5 seeds. Across all eight hyperparameters, the curves are smooth and mostly plateau-shaped, indicating that FedSSM does not rely on narrow hyperparameter tuning.

### B.2. Hyperparameter Sensitivity

We further evaluate the sensitivity of FedSSM to all major hyperparameters, including the budget range, exploration and exploitation coefficients, mixing sharpness, trust penalty, and budget response strength. Figure 6 reports one-at-a-time sensitivity results on VQA v2.0 and UCF-101 under $\alpha = 0.1$, 100 clients, 50% partial-modality, and 5 random seeds.

Figure 6 shows that FedSSM is stable across a reasonably wide range of settings. The budget parameters $N_{\min}$ and $N_{\max}$ have broad interior optima, suggesting that the adaptive budget mechanism is not tied to a single participation size. The coefficients $\beta_1$, $\beta_2$, and $\beta_3$ show smooth trends, indicating that the exploration and exploitation scores tolerate moderate perturbations. The parameters $\tau$, $\eta$, and $\kappa$ also exhibit mild variations without sharp performance cliffs.

These trends are consistent across VQA v2.0 and UCF-101, which suggests that the hyperparameters have transferable roles rather than dataset-specific effects. We attribute this stability to the self-correcting nature of the surprise signal: suboptimal settings may change the aggressiveness of exploration or budgeting, but the resulting prediction error increases $S_t$, which in turn triggers more exploration and more participation.

### B.3. Fair Comparison under Equal Participation

A possible concern is that FedSSM may benefit from using fewer or more clients per round rather than from better client selection. To isolate the effect of selection quality, we compare all methods under the same fixed 10% participation budget on VQA v2.0. We also report FedAvg with 9% and 15% participation to test whether naively changing the number of selected clients can close the gap.

Table 8 shows that FedSSM remains stronger than all baselines even when its participation budget is fixed to the same 10% level. Therefore, the main improvement does not come from simply selecting fewer clients. The adaptive version further improves both accuracy and convergence speed, showing that adaptive budgeting is beneficial on top of the predictive selection mechanism.

### B.4. Dataset-Level Heterogeneity

The main experiments use Dirichlet-based splits to simulate label imbalance. To further test realistic heterogeneity, we evaluate a cross-dataset protocol where VQA v2.0, Hateful Memes, and GQA are assigned to separate client groups. Each group observes only its own dataset distribution while sharing the image-text modality pair. We compare FedSSM with representative client selection and multimodal FL baselines, including Oort (Lai et al., 2021), Grace (Zhang et al.,

*Table 8.* Fair computational comparison on VQA v2.0 under $\alpha = 0.1$. All methods use fixed 10% participation unless otherwise specified. Target time is computed as time per round multiplied by the rounds needed to reach 65% accuracy. "–" means the method does not reach the target within 100 rounds.

| Method | Participation | Acc. (%) | Time/Round (s) | Target Time (s) |
| --- | --- | --- | --- | --- |
| FedAvg | 9% fixed | 57.53 | 47.1 | – |
| FedAvg | 10% fixed | 58.42 | 51.8 | – |
| FedAvg | 15% fixed | 59.84 | 76.2 | – |
| FedMBridge | 10% fixed | 65.23 | 70.5 | 3666.0 |
| Grace | 10% fixed | 64.82 | 55.8 | – |
| FedDAT | 10% fixed | 65.71 | 62.3 | 2990.4 |
| Co-LoRA | 10% fixed | 65.94 | 64.1 | 2948.6 |
| LRR | 10% fixed | 66.23 | 58.4 | 2511.2 |
| FedSSM | 10% fixed | **67.15** | 52.3 | **1464.4** |
| FedSSM | $\sim$9% adaptive | **68.47** | **45.2** | **994.4** |

*Table 9.* Dataset-level heterogeneity results. VQA v2.0, Hateful Memes, and GQA are assigned to separate client groups. $\Delta_{\mathrm{LRR}}$ denotes the improvement of FedSSM over the strongest baseline LRR.

| Method | Global Acc. (%) | Worst-group Acc. (%) | Rounds | Round Saving |
| --- | --- | --- | --- | --- |
| FedAvg | 49.72±0.81 | 41.23±1.12 | 78±5 | – |
| Oort | 51.34±0.73 | 43.47±1.04 | 68±4 | – |
| FedMBridge | 54.14±0.72 | 47.35±0.97 | 58±4 | – |
| Grace | 54.52±0.67 | 47.73±0.93 | 55±3 | – |
| FedDAT | 55.28±0.62 | 48.64±0.87 | 52±3 | – |
| Co-LoRA | 55.51±0.61 | 48.92±0.85 | 50±3 | – |
| LRR | 55.83±0.58 | 49.37±0.82 | 48±3 | 0.0% |
| FedSSM | **56.74±0.55** | **50.43±0.76** | **41±3** | **14.6%** |

2025), FedDAT (Chen et al., 2024), Co-LoRA (Seo et al.), and LRR (Nguyen et al., 2026). FedDAT targets heterogeneous multimodal foundation-model finetuning, Co-LoRA studies collaborative personalization on heterogeneous multimodal clients, and LRR addresses missing-data representation learning in multimodal FL. These methods cover complementary directions to our selection-centric framework.

The cross-dataset results show that FedSSM is not limited to synthetic Dirichlet heterogeneity. It improves both global accuracy and worst-group accuracy while using fewer communication rounds. This suggests that proactive selection can better balance client groups whose data distributions differ at the dataset level.

### B.5. Three-Modality Benchmark

To verify that FedSSM is not restricted to two-modality benchmarks, we additionally evaluate it on CMU-MOSEI (Zadeh et al., 2018), which contains text, audio, and video modalities. We report Acc-7, Macro F1, and communication rounds under 100 clients, 50% partial-modality, and $\alpha = 0.1$.

FedSSM achieves the best Acc-7 and Macro F1 while requiring fewer rounds. This result indicates that the predictive selection mechanism remains effective when the modality space expands beyond two modalities.

### B.6. Scalability and Robustness Analysis

We conduct a comprehensive evaluation of scalability and robustness across UCF-101, Hateful Memes, and MiT-51 benchmarks. Figure 7 presents the performance trajectories under varying system configurations.

**Scalability.** We vary the total number of clients from 20 to 200 while maintaining a constant participation ratio. As shown in the left panels of Figure 7, FedSSM consistently outperforms baseline methods in both accuracy and communication efficiency. On Hateful Memes, FedSSM maintains high AUC-ROC even with 200 clients, whereas reactive methods such as FedAvg and mmFedMC degrade more clearly. The efficiency plots further show that FedSSM requires fewer communication rounds to reach the target accuracy. This indicates that the counterfactual selection mechanism remains effective as the

*Table 10.* Results on the three-modality CMU-MOSEI benchmark. The dataset contains text, audio, and video modalities. Gain and round saving are computed against FedMBridge.

| Method | Acc-7 | Macro F1 | Rounds | F1 Gain | Round Saving |
|---|---|---|---|---|---|
| FedAvg | 40.31±0.92 | 34.18±1.08 | 52±5 | – | – |
| Oort | 42.15±0.84 | 36.42±0.96 | 47±4 | – | – |
| CreamFL | 43.68±0.79 | 37.95±0.91 | 41±4 | – | – |
| mmFedMC | 44.21±0.75 | 38.53±0.87 | 39±4 | – | – |
| FedMBridge | 44.85±0.71 | 39.27±0.82 | 36±3 | 0.00 | 0.0% |
| FedSSM | **48.23±0.63** | **42.61±0.68** | **24±3** | **+3.34** | **33.3%** |

client population grows.

**Robustness against Missing Modalities.** We simulate severe modality heterogeneity by increasing the ratio of partial-modality clients from 0% to 75%. The right panels of Figure 7 show that FedSSM is more robust than existing methods under missing modalities. On MiT-51, the performance gap between FedSSM and the baselines becomes larger as the missing ratio increases. This supports the role of surprise-calibrated aggregation: when the predicted trajectory becomes unreliable, the server reduces the influence of updates that may destabilize multimodal fusion.

## C. Implementation Details

This appendix provides detailed implementation specifications that complement the main algorithm presented in Section 3.

### C.1. Detailed Algorithm

Algorithm 2 expands upon Algorithm 1 with explicit handling of online adaptation, stale client statistics, and the warm-up procedure.

### C.2. Local Training and Reproducibility Details

All methods share the same local training protocol unless otherwise specified. Each selected client performs 5 local epochs using SGD with learning rate 0.01, momentum 0.9, weight decay $5 \times 10^{-4}$, and batch size 64. The global model architecture, modality encoders, fusion module, data partitions, and random seeds are kept identical across methods. We use 5 random seeds for all reported results.

FedSSM introduces a small server-side SSM module. This module is optimized with Adam using learning rate $10^{-3}$, $\beta_1 = 0.9$, $\beta_2 = 0.999$, and weight decay $10^{-5}$. This optimizer is only used for the server-side SSM predictor, not for local model training. Baselines without trainable server-side predictors therefore do not require an additional server optimizer. This distinction avoids changing the local optimization protocol and ensures that the comparison focuses on client selection rather than different local training dynamics.

For hyperparameter tuning, we use a held-out validation split and tune each baseline according to its recommended search space. FedSSM hyperparameters are selected on the validation set and then fixed across all test runs. The same validation budget is used for all methods.

### C.3. Stale Client Statistics Estimation

For clients not selected in recent rounds, we estimate their loss variance using temporal decay:

$$v_i^{(t)} = \gamma^{\Delta t_i} v_i^{(t-\Delta t_i)} + (1 - \gamma^{\Delta t_i})\bar{v}, \tag{32}$$

where $\gamma \in (0,1)$ is the decay factor and $\bar{v}$ denotes the global mean variance across all clients. As staleness $\Delta t_i$ increases, estimates regress toward the population mean, reflecting increased uncertainty about outdated statistics. This design complements the exploration weight $w_i^{\text{explore}}$ in Eq. (11), which naturally encourages sampling stale clients to refresh their statistics.

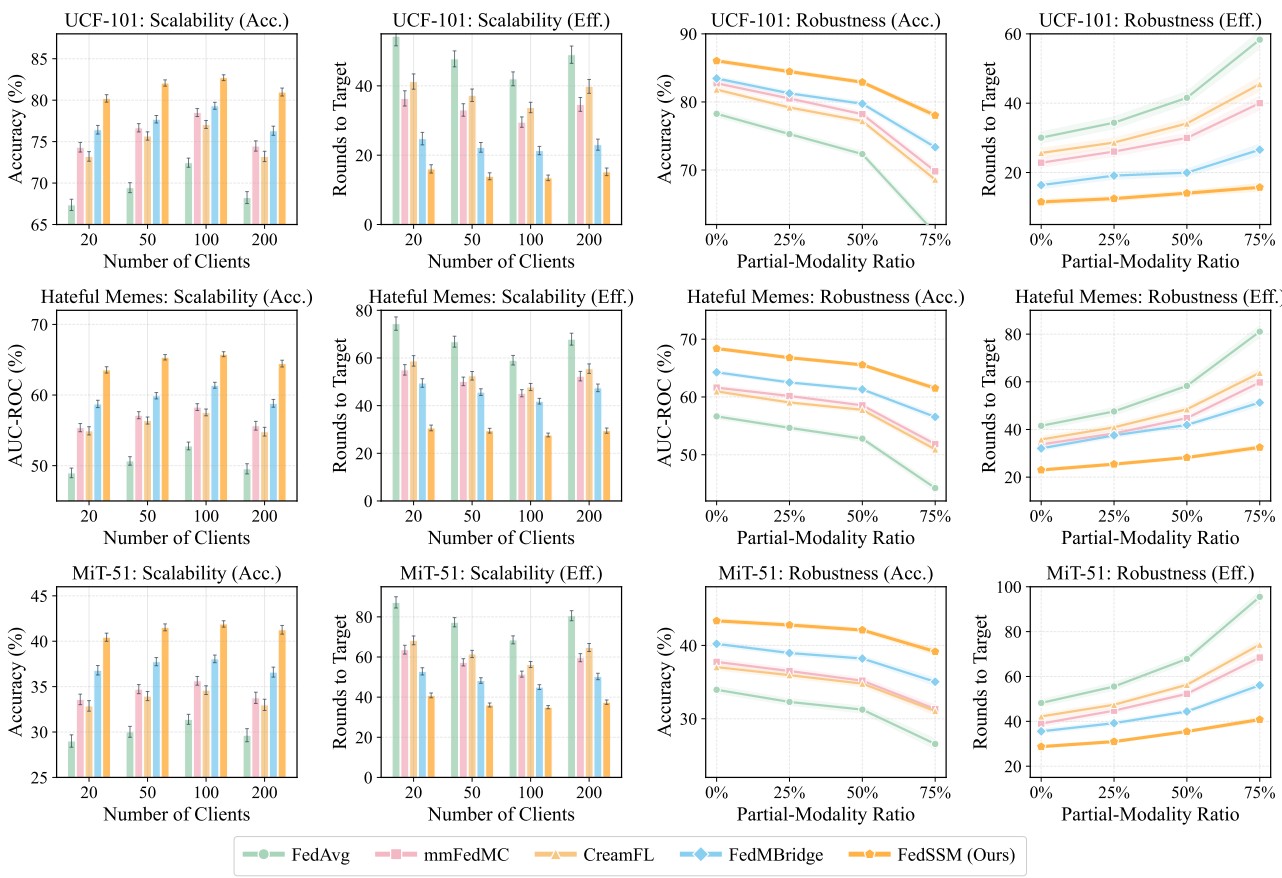

*Figure 7.* Scalability and robustness analysis. The left panels report performance under increasing client numbers. The right panels report robustness under varying partial-modality ratios.

## C.4. Adaptive Threshold Update

The surprise threshold $\theta_S$ in Eq. (10) adapts online via exponential moving average:

$$\theta_S^{(t)} = \beta\theta_S^{(t-1)} + (1 - \beta)S_t, \tag{33}$$

where $\beta$ is the smoothing factor consistent with other components. This allows $\theta_S$ to track the evolving baseline of prediction uncertainty throughout training.

## C.5. Action Space Design

The action space is defined as $\mathcal{A} = \mathcal{N} \times \mathcal{P}$, where:

- $\mathcal{N} = \{N_{\min}, N_{\min} + \Delta N, \ldots, N_{\max}\}$ discretizes the budget range with step size $\Delta N = 2$. With $N_{\min} = 5$ and $N_{\max} = 15$, this yields $|\mathcal{N}| = 6$ candidate budgets.

- $\mathcal{P} = \{0.0, 0.25, 0.5, 0.75, 1.0\}$ contains five mixing coefficients that interpolate between pure exploration ($\rho = 0$) and pure exploitation ($\rho = 1$) in Eq. (11).

The total action space size is $|\mathcal{A}| = |\mathcal{N}| \times |\mathcal{P}| = 6 \times 5 = 30$, independent of the client population $K$. For each candidate action $\mathbf{a} \in \mathcal{A}$, counterfactual evaluation via Eq. (8) requires $O(d^2)$ operations for the state transition, yielding total complexity $O(|\mathcal{A}| \cdot d^2) = O(30 \cdot 64^2) \approx 1.2 \times 10^5$ operations per round. This is negligible compared to local training and communication overhead.

---

**Algorithm 2** FedSSM: Detailed Implementation with Online Adaptation

---

**Require:** Clients $[K]$, datasets $\{\mathcal{D}_i\}$, initial model $\boldsymbol{\theta}_0$, total rounds $T$, warm-up rounds $T_w$

1: **Initialize:** SSM parameters ($\mathbf{A}_{\text{base}}$ via HiPPO, $\mathbf{B}_{\text{base}}$, $\mathbf{C}$); latent state $\mathbf{h}_0 \leftarrow \mathbf{0}$; threshold $\theta_S \leftarrow \theta_S^{(0)}$
2: **Initialize:** Client statistics $\{v_i \leftarrow 0, \Delta t_i \leftarrow 0\}_{i=1}^K$
3: **for** $t = 1, \ldots, T$ **do**
4:     `// Phase 1:  State Space Prediction`
5:     Compute $\mathbf{h}_t, \hat{\mathbf{x}}_t$ via Eq. (5)-(6)
6:     `// Phase 2:  Client Selection`
7:     **if** $t \leq T_w$ **then**
8:         $N_t \leftarrow \lfloor (N_{\min} + N_{\max})/2 \rfloor$; select $N_t$ clients uniformly at random
9:     **else**
10:        Update stale statistics via Eq. (32) for non-selected clients
11:        Compute $N_t$ via Eq. (10) using $S_{t-1}$
12:        Compute selection probabilities $\{q_i^{(t)}\}$ via Eq. (11)
13:        Sample $N_t$ clients according to $\{q_i^{(t)}\}$
14:     **end if**
15:     `// Phase 3:  Distributed Local Training`
16:     **for** each selected client $i$ **in parallel do**
17:        $\boldsymbol{\theta}_{i,t} \leftarrow \text{LocalUpdate}(\boldsymbol{\theta}_t, \mathcal{D}_i, \mathcal{M}_i)$
18:        Compute and upload: local loss $L_i$, variance $v_i$
19:        $\Delta t_i \leftarrow 0$                       ▷ Reset staleness
20:     **end for**
21:     `// Phase 4:  Observation and Online Update`
22:     Aggregate $\mathbf{x}_t \leftarrow [L_t, \|\nabla L_t\|, \sigma_t^2, t/T]^\top$
23:     Compute surprise $S_t$ via Eq. (7)
24:     Update SSM: $\min \|\mathbf{x}_t - \hat{\mathbf{x}}_t\|^2$ with gradient clipping
25:     Update threshold: $\theta_S \leftarrow \beta \theta_S + (1 - \beta) S_t$
26:     `// Phase 5:  Trust-Weighted Aggregation`
27:     Compute $\{\tau_i\}$ via Eq. (13)
28:     Update $\boldsymbol{\theta}_{t+1}^m$ via Eq. (12); $\boldsymbol{\theta}_{t+1}^{\text{fus}}$ via Eq. (14)
29:     $\Delta t_i \leftarrow \Delta t_i + 1$ for all non-selected clients
30: **end for**
31: **return** $\boldsymbol{\theta}_T$

---

In practice, we observe that the adaptive formulas in Eqs. (10) and (11) closely approximate the utility-maximizing action from $\mathcal{A}$. We therefore use the closed-form solutions for efficiency while retaining the counterfactual framework for conceptual clarity and potential extensions to more complex action spaces.

### C.6. Online Training Stability

The SSM parameters are updated online by minimizing the prediction error $\|\mathbf{x}_t - \hat{\mathbf{x}}_t\|^2$. At each communication round $t$, the server performs a single gradient step using Adam optimizer with learning rate $10^{-3}$, $\beta_1 = 0.9$, $\beta_2 = 0.999$, and weight decay $10^{-5}$. The loss function incorporates a forgetting factor $\lambda = 0.95$ to prioritize recent observations:

$$\mathcal{L}_{\text{SSM}}^{(t)} = \sum_{\tau=1}^t \lambda^{t-\tau} \|\mathbf{x}_\tau - \hat{\mathbf{x}}_\tau\|^2. \tag{34}$$

In practice, we maintain a running estimate of this loss using exponential moving averages rather than storing the full history.

To ensure stability, we employ three mechanisms: (1) gradient clipping with max norm 1.0 to prevent exploding updates; (2) the forgetting factor $\lambda$ that enables adaptation to non-stationary training dynamics by down-weighting distant observations; and (3) a warm-up period of $T_w = 10$ rounds during which random selection is used to accumulate sufficient observations before activating proactive inference. During warm-up, the SSM is trained but its predictions are not used for client selection.

The SSM parameters ($\mathbf{A}_{\text{base}}$, $\mathbf{B}_{\text{base}}$, $\mathbf{C}$, $\mathbf{W}_A$, $\mathbf{W}_B$, $\mathbf{W}_N$, $\mathbf{W}_\pi$) are initialized using Xavier initialization, except $\mathbf{A}_{\text{base}}$ which is initialized via HiPPO (Gu et al., 2020) for stable long-range dependency modeling.

### C.7. Computational Complexity and Scalability

Table 11 reports the computational overhead with varying client populations. The SSM state update and counterfactual reasoning operate on fixed-dimensional latent states, yielding $O(d^2)$ and $O(|\mathcal{A}| \cdot d^2)$ complexity respectively, both independent of $K$. Only the client selection step scales linearly at $O(K)$ for computing selection probabilities. As shown in the table, total overhead increases marginally as $K$ grows, confirming that FedSSM scales efficiently to larger client populations.

*Table 11.* Computational overhead analysis with varying client populations.

| Component | Time (s) | | | | Memory |
|---|---|---|---|---|---|
| | $K$=50 | $K$=100 | $K$=150 | $K$=200 | (MB) |
| SSM State Update | 0.30 | 0.30 | 0.30 | 0.30 | 1.5 |
| Counterfactual Reasoning | 0.40 | 0.40 | 0.40 | 0.40 | 0.6 |
| Client Selection | 0.06 | 0.10 | 0.14 | 0.18 | 0.2 |
| **Total Overhead** | **0.76** | **0.80** | **0.84** | **0.88** | **2.3** |

## D. Limitations

FedSSM still has several limitations. First, although the sensitivity analysis shows stable performance across a broad range of settings, the method introduces more hyperparameters than purely reactive client selection methods, and a small validation set may be needed when transferring it to new scenarios. Second, our experiments mainly consider simulated or controlled client populations. FedSSM can mitigate stale or newly joined clients through population-mean initialization and stale-statistics regression, but highly dynamic cross-device environments deserve more systematic study. Third, the SSM models aggregate training dynamics rather than fine-grained client-level behavior, which keeps the method lightweight but may miss subgroup-specific changes. Extending FedSSM to asynchronous training and more complex real-world multimodal systems is an important direction for future work.

