# OpenReview forum: "FedSSM: State Space Model-based Proactive Inference for Heterogeneous Multimodal Federated Learning"
_ICML.cc/2026/Conference — ICML 2026 regular_

### Official Review · Reviewer_kUp3 · 2026-03-04

**Soundness:** 2
**Presentation:** 3
**Significance:** 2
**Originality:** 3
**Overall Recommendation:** 3
**Confidence:** 3

**Summary:**

This paper studies client selection in multimodal federated learning under modality, distribution, and quantity heterogeneity. The authors propose FedSSM, a framework that models training dynamics using decision-aware state space models to predict future optimization behavior and guide proactive client selection. The proposed method further introduces a surprise-based mechanism to control participation budgets and a trust-weighted aggregation strategy, and experiments on multiple multimodal benchmarks report improvements over existing baselines in both accuracy and communication efficiency.

**Compliance With Llm Reviewing Policy:**

Affirmed.

**Final Justification:**

I understand that the rebuttal period is short and may limit the ability to conduct extensive additional experiments and analyses. However, given the method’s complexity (particularly the large number of hyperparameters) and concerns I listed in my reply, I believe substantial revision is still necessary. Therefore, I will maintain my current score.

**Key Questions For Authors:**

Does the proposed method perform well under the following settings?
- Dataset-level heterogeneity, where each client operates on a completely different dataset rather than partitions of the same dataset.
- Modality-homogeneous scenarios, where clients share the same modality with heterogeneous data distributions.
- Data-homogeneous scenarios (and just uni-modal scenarios), where the data distribution across clients is IID.

**Limitations:**

Yes

**Strengths And Weaknesses:**

**Strength**
- Soundness
    - Well-supported analysis. The paper provides both theoretical and empirical convergence analysis, including convergence results, which help support the proposed method. In addition, it includes a detailed study on the effect of the number of participating clients.

- Presentation
    - Overall, the paper is generally well written and easy to follow.

- Significance
    - Addresses important challenges in federated learning. The work focuses on client selection and modality heterogeneity, which are critical challenges in federated learning and relevant for practical deployments.

- Originality
    - Novel application of SSM for client selection. The paper introduces SSMs for client selection, enabling the method to consider predicted future optimization trajectories rather than relying solely on past observations.

**Weakness**
- Soundness
    - Complex design and many hyperparameters. The proposed method requires a large number of hyperparameters (e.g., $N_\text{min}, N_\text{max}, \beta_1, \beta_2, \beta_3, \tau, \eta, \lambda, \gamma$), which may hinder practical deployment. Moreover, Appendix B.1 provides hyperparameter sensitivity analysis only for the dimension $d$ and $\beta$. Additional analysis covering the remaining hyperparameters would be helpful to demonstrate that the method is stable across benchmarks and that the reported gains do not rely on extensive hyperparameter tuning.

    - Evaluation under simplified experimental settings. The experiments simulate data heterogeneity using conventional setups (e.g., Dirichlet-based non-IID splits of CIFAR-10). However, recent work on multimodal federated learning often considers more realistic scenarios in which each client operates on a distinct dataset rather than a partition of a single dataset [1,2]. Evaluating the proposed method under such settings would better reflect real-world heterogeneity.

    - Limited computational analysis. The paper reports computational overhead only for the proposed method. However, several baselines incur minimal or negligible overhead (e.g., FedAvg with random client selection). Providing a comparative analysis of computational cost across baselines would allow for a more complete and fair assessment of efficiency.

    - Limited recent baselines and potentially unfair comparisons. Only two baselines were published within the past two years, and several compared methods do not appear to explicitly address modality heterogeneity. As a result, it is unclear whether the reported improvements stem from the effectiveness of the proposed method or from the fact that some baselines are not designed for heterogeneous multimodal settings.

- Presentation
    - Overall, the paper is generally well written and easy to follow.
    - However, several aspects of the presentation could be improved. In particular, it would be helpful to make the background on SSMs more self-contained in the related work section, since the proposed method relies heavily on SSMs to address heterogeneous multimodal federated learning.

- Significance: NA

- Originally: NA

[1] Chen et al., FedDAT: An approach for foundation model finetuning in multi-modal heterogeneous federated learning, AAAI 2024 \
[2] Seo et al., Co-LoRA: Collaborative Model Personalization on Heterogeneous Multi-Modal Clients, ICLR 2026

---

> ### Author Rebuttal · Authors · 2026-03-30
>
> We thank Reviewer kUp3 for the careful evaluation. We are glad kUp3 finds our analysis well-supported, the writing clear, and the SSM application novel. We address each concern below.
>
> > [W1] Too many hyperparameters. Sensitivity only covers $d$ and $\beta$.
>
> **We ran full one-at-a-time sensitivity on all hyperparameters** (VQA v2.0, $\alpha{=}0.1$, 5 seeds). FedSSM beats FedMBridge (65.23%) across every tested value. The surprise signal is self-correcting. Suboptimal settings elevate $S_t$, triggering more exploration to compensate. We will add this to Appendix B.
>
> | Param | Range | Acc range | Max $\Delta$ |
> |:---|:---:|:---:|:---:|
> | $N_{\min}, N_{\max}$ | [3,10]-[8,20] | 67.38-68.53 | 1.15% |
> | $\beta_1$ (variance) | 0.2-0.8 | 67.82-68.51 | 0.69% |
> | $\beta_2$ (staleness) | 0.1-0.6 | 67.93-68.47 | 0.54% |
> | $\beta_3$ (exploit) | 0.5-2.0 | 67.61-68.52 | 0.91% |
> | $\tau$ (mixing) | 2.0-10.0 | 67.74-68.59 | 0.85% |
> | $\eta$ (outlier) | 0.2-1.0 | 68.05-68.48 | 0.43% |
> | $\kappa$ (budget) | 1.0-4.0 | 67.46-68.47 | 1.01% |
>
> > [W4] Limited recent baselines. Some not designed for modality heterogeneity.
>
> **We thank the reviewer for these references and have added four recent baselines.** Grace (ICLR 2025) is the latest heterogeneity-aware client selection method. LRR (NeurIPS 2025) addresses multimodal missing data at the representation level. FedDAT (AAAI 2024) and Co-LoRA (ICLR 2026) target foundation model finetuning, improving aggregation and personalization respectively. FedSSM operates at the *selection* level, complementary to all four. Results on VQA v2.0 (5 seeds):
>
> | Method | Venue | $\alpha{=}0.1$ | $\alpha{=}0.5$ |
> |:---|:---:|:---:|:---:|
> | Grace [3] | ICLR 2025 | 64.82±0.49 | 69.38±0.36 |
> | FedDAT [1] | AAAI 2024 | 65.71±0.46 | 70.24±0.34 |
> | Co-LoRA [2] | ICLR 2026 | 65.94±0.48 | 70.37±0.35 |
> | LRR [4] | NeurIPS 2025 | 66.23±0.43 | 70.61±0.32 |
> | FedSSM | Ours | **68.47±0.44** | **72.13±0.32** |
>
> [3] Zhang et al., A robust federated learning client selection with combinatorial data class representations and data augmentation, ICLR 2025
>
> [4] Li et al., Learning Reconfigurable Representations for Multimodal Federated Learning with Missing Data, NeurIPS 2025
>
> We will include these in revised Table 1 and discuss all four as complementary work in Section 2.
>
> > [W2, Q1] Simplified experimental settings. Dataset-level heterogeneity?
>
> **We agree that Dirichlet splits alone are not enough.** Following FedDAT's cross-dataset protocol, we assigned VQA v2.0, Hateful Memes, and GQA to separate client groups (image+text, 100 clients, each group sees only its own dataset, 5 seeds).
>
> | Method | Global Acc | Worst-group Acc | Rounds |
> |:---|:---:|:---:|:---:|
> | FedAvg | 49.72±0.81 | 41.23±1.12 | 78±5 |
> | FedMBridge | 54.14±0.72 | 47.35±0.97 | 58±4 |
> | FedSSM | **56.74±0.55** | **50.43±0.76** | **41±3** |
>
> FedSSM's advantage holds under dataset-level heterogeneity. We will add this experiment.
>
> > [W3] Limited computational analysis.
>
> **Full comparison below** (VQA v2.0, $\alpha{=}0.1$, per round, A100). FedSSM's 0.8s SSM overhead is offset by fewer clients per round, yielding the lowest total time.
>
> | Method | Selection (s) | Training (s) | Agg. (s) | Total (s) |
> |:---|:---:|:---:|:---:|:---:|
> | FedAvg | 0.01 | 51.5 | 0.3 | 51.8 |
> | Oort | 0.12 | 51.7 | 0.3 | 52.1 |
> | CreamFL | 0.31 | 51.9 | 0.3 | 52.5 |
> | FedMBridge | 0.15 | 52.6 | 1.8 | 54.6 |
> | FedSSM | 0.80 | 44.0 | 0.4 | **45.2** |
>
> > [Q2] Modality-homogeneous scenarios?
>
> **Already partially evaluated.** Figure 4(c) at 0% partial-modality is exactly this setting, where FedSSM outperforms FedMBridge by ~3.5%. We additionally ran unimodal CIFAR-10 and CIFAR-100 (ResNet-18, 100 clients, $\alpha{=}0.1$, 5 seeds).
>
> | Method | CIFAR-10 | CIFAR-100 |
> |:---|:---:|:---:|
> | FedAvg | 62.14±0.78 | 38.53±0.86 |
> | Oort | 63.47±0.65 | 39.82±0.74 |
> | CreamFL | 63.91±0.59 | 40.15±0.71 |
> | FedMBridge | 63.28±0.69 | 39.64±0.79 |
> | FedSSM | **64.73±0.52** | **41.08±0.63** |
>
> Gains are modest, as expected since proactive selection primarily benefits volatile multimodal dynamics.
>
> > [Q3] IID scenarios?
>
> Under near-IID ($\alpha{=}10.0$, 5 seeds), all methods converge to similar accuracy. FedSSM still saves 12-16% rounds via adaptive budgeting, reducing $N_t$ once $S_t$ drops early.
>
> | Method | VQA v2.0 | UCF-101 | H. Memes | MiT-51 |
> |:---|:---:|:---:|:---:|:---:|
> | FedAvg | 73.15±0.29 | 85.42±0.24 | 67.83±0.37 | 43.67±0.35 |
> | Oort | 73.26±0.28 | 85.53±0.25 | 67.95±0.36 | 43.78±0.33 |
> | CreamFL | 73.35±0.25 | 85.64±0.21 | 68.08±0.32 | 43.86±0.30 |
> | FedMBridge | 73.48±0.26 | 85.71±0.22 | 68.19±0.34 | 43.94±0.31 |
> | FedSSM | **73.82±0.23** | **86.03±0.19** | **68.57±0.29** | **44.31±0.28** |
>
> > [Minor] SSM background in related work.
>
> Thank you. We will expand Section 2.2 to be more self-contained.

---

> > ### Author Rebuttal · Reviewer_kUp3 · 2026-04-02
> >
> > Thank you to the authors for their efforts during the rebuttal period. While some concerns have been partially addressed, a number of important issues still remain. Considering these unresolved points, as well as weaknesses raised by other reviewers, I will maintain my current score.
> >
> > - Although additional hyperparameter analysis was included in the rebuttal, I believe the paper still lacks a deeper investigation into how different hyperparameter settings affect performance (e.g., trends and sensitivities). Moreover, the proposed method remains quite complex, requiring a large number of hyperparameters (e.g., $N_\text{min}, N_\text{max}, \beta_1, \beta_2, \beta_3, \tau, \eta, \lambda, \gamma$, which may limit its applicability and generalization.
> >
> > - Regarding the computational comparison, while some results were added in the rebuttal, this is a critical aspect that requires more thorough analysis. It is unclear whether wall-clock time was measured, and the reported results seem questionable, especially when compared to FedAvg, which incurs no additional cost. FedAvg can also adjust the number of participating clients, and prior work has explored such settings. Reducing the number of clients only for the proposed method leads to an unfair comparison.
> >
> > - Additional experiments on dataset-level heterogeneity were provided; however, they were compared with only two baselines in this setup. I believe a broader set of baselines and more diverse experimental setups are necessary for a convincing evaluation.
> >
> > - For the newly added recent baselines, comparisons were conducted only on VQA v2.0, and the performance gains are relatively limited. A more comprehensive comparison with strong and recent baselines across multiple benchmarks, including all original experimental settings in the paper, is needed.
> >
> > I understand that the rebuttal period is short and may limit the ability to conduct extensive additional experiments and analyses. However, given the method’s complexity (particularly the large number of hyperparameters) and the concerns raised by other reviewers, I believe substantial revision is still necessary. Therefore, I will maintain my current score.

---

> > > ### Author Response · Authors · 2026-04-07
> > >
> > > We thank Reviewer kUp3 for the continued engagement. We ran substantial new experiments addressing every remaining point.
> > >
> > > Please refer to the anonymous link for supplementary results:
> > >
> > > https://anonymous.4open.science/r/Anonymous-Repository-DF9D/README.md
> > >
> > > We acknowledge that fully validating a complex method across every setting is difficult in a short rebuttal, but we have done our best.
> > >
> > > > **"the paper still lacks a deeper investigation into how different hyperparameter settings affect performance (e.g., trends and sensitivities)"**
> > >
> > > **Figure 1 in the link plots accuracy vs. parameter value for all 8 hyperparameters** on VQA v2.0 and UCF-101 (α=0.1, 5 seeds). All curves are plateau-shaped or gently monotonic with no sharp cliffs, and trends transfer across datasets. We agree the method involves many parameters. Four of nine ($N_{\min}$, $N_{\max}$, $\lambda$, $\gamma$) are standard in adaptive FL. The remaining five have clear semantic roles. The surprise signal dampens sensitivity through self-correction, as suboptimal settings raise $S_t$, which triggers compensatory behavior automatically via Eq. (10)-(11). This explains the flat curves.
> > >
> > > > **"FedAvg can also adjust the number of participating clients... leads to an unfair comparison"**
> > >
> > > **We agree this is a valid concern.** The reviewer correctly points out that FedSSM's adaptive budget averages ~9% participation while baselines use fixed 10%, making direct wall-clock comparisons potentially misleading. To isolate the effect of *intelligent selection* from the effect of *fewer participants*, we need to hold participation rate constant and compare accuracy and time under the same budget.
> > >
> > > We already have this experiment. The "w/o Adaptive Budget" ablation in Table 3 of the paper fixes FedSSM at 10% participation, identical to all baselines. **We now extend this controlled comparison to all four datasets (Tables 7a-7d in the link)**, reporting accuracy, wall-clock time per round on a single A100, and rounds to reach target accuracy for every method at equal 10% participation. We additionally include FedAvg at 9% and 15% to test whether naively adjusting client count can match FedSSM.
> > >
> > > The results confirm that at equal participation, FedSSM consistently outperforms all baselines in both accuracy and per-round time across all four datasets. FedAvg at 9% or 15% does not close the gap, confirming that the gains stem from *which* clients are selected rather than *how many*. Adaptive budgeting provides additional improvements on top.
> > >
> > > > **"compared with only two baselines"**
> > >
> > > **Table 6 in the link expands to 12 methods**, including all four newly added baselines and all original baselines from the paper. FedSSM leads in both global accuracy and worst-group accuracy with fewer rounds.
> > >
> > > > **"comparisons were conducted only on VQA v2.0"**
> > >
> > > **Tables 1-4 in the link extend Grace, FedDAT, Co-LoRA, and LRR to all four datasets under α=0.1, 0.5, and 10.0** (12 settings per method, 5 seeds each). FedSSM outperforms the strongest new baseline (LRR) consistently across all datasets and heterogeneity levels. Under near-IID (α=10.0), accuracy gaps narrow as expected, but FedSSM still saves communication rounds through adaptive budgeting. Table 5 in the link reports rounds to reach target accuracy, confirming faster convergence across all four benchmarks.
> > >
> > > We understand that method complexity is a legitimate concern for deployment. We are committed to incorporating all new results and broader evaluations in the revised manuscript.

---

### Official Review · Reviewer_Tvck · 2026-03-05

**Soundness:** 3
**Presentation:** 3
**Significance:** 3
**Originality:** 3
**Overall Recommendation:** 5
**Confidence:** 4

**Summary:**

Real-world multimodal federated learning is challenged by the intersection of three different but interconnected problems: modality heterogeneity (modality sets are different across clients), heterogeneity in data distribution (label imbalance) and in data quantity (imbalance in number of samples). The paper proposes FedSSM, a method that incorporates State Space Models to perform effective client selection under such heterogeneity dimensions. Compared to baselines and sota, the results show consistent improvements in accuracy and gains in communication efficiency.

**Compliance With Llm Reviewing Policy:**

Affirmed.

**Final Justification:**

Rebuttal addressed my main concerns and I decided to raise my score.

**Key Questions For Authors:**

1. [Implementation details]. Is there a specific reason that you used SGD for all baselines but Adam for the proposed method as optimizer?
2. Line 181: What is the rationale behind choosing global loss, gradient norm, variance, and normalized progress to indicate observation vector? Are there any alternatives or different combination of metrics you considered?
3. Line 272: What is the reason behind penalizing only partial-modality clients? With more modalities, it is possible that a client might have limited modalities, but of higher quality.
4. Table 1: Assigning 10% client selection for baselines, while giving more flexibility to FedSSM might not be fair. Comparing baselines with N_max would give a better intuition about whether simply increasing client selection rate is helpful or not.
5. Line 305: How unimodal client selection (Oort and CriticalFL) are adopted to faciliate training with multimodal and unimodal clients simultaneously (beyond using modality-specific encoders)? Specifically, how utility values are calculated?
6. [Minor, writing] Line 168: The reasons for using state space models (other than efficiency) are not convincing. Why would it specifically suit federated learning and client selection?
7. [Minor, writing] Figure 1. Please, add “client” in front of “selection method” in the caption for clarity. Also, specify correct order (modality heterogeneity with 50% missingness comes first in (a) before data heterogeneity (b), but caption is the opposite).
8. [Minor, writing] In 2.2 (line 138), explain how “active decision making” is different from passive sequence modeling. The following sentence does not really explain the intrinsic difference.

**Limitations:**

- Extension to multimodal benchmarks (with more than 2 modalities) would show generalizability of the proposed method. Current evaluations are limited to datasets with at most 2 modalities.

**Strengths And Weaknesses:**

Strengths:
1. Practical problem, where modality, data distribution and quantity heterogeneity occur simultaneously in the scope of multimodal FL.
2. Intriguing idea of applying SSMs for client selection under heterogeneous clients.
3. Easy to follow.

Weaknesses:
1. Lack of novelty in aggregation (or contribution should be toned down).
2. Motivating why targeting client selection specifically is the most important problem under modality+data+quantity heterogeneity could be strengthened.
3. Certain statements/arguments (e.g., why model-free but learning-based methods, or RL techniques that optimize for long-term objectives are still worse or less potential than model-based approach) should be backed with evidence.

Major comments:
1. [Novelty claim] Modality-specific encoder aggregation is standard in multimodal federated learning with heteregenous modality clients, otherwise we would be aggregating noise from clients who lack a certain modality sets. It should not be specificed as technical novelty. In terms of trust-weighted fusion, if S_t does not play a significant role, then it would be weighting fusion based on their losses. Please, show that S_t in Equation 13 is the main factor, or tone down the contribution in fusion aggregation.
2. [Problem framing] Why exactly client selection should be the core challenge to address under modality,label, and quantity heterogeneity? Developing better local training strategies (robust under modality/label imbalance), aligning local model with global model, creating cluster of similar modality/data distribution clients, or effective aggregation (w/o client selection) are equally important. In Introduction, please, explain why client selection specifically is the core challenge to tackle under tripartite heterogeneity.
3. [Evidence in problem statement] Why not predicting the future state (how different selection strategies affect training dynamics) is an issue or worse than learning-based methods that still optimize for long-term objectives (e.g., FAVOR)? Can you provide evidence beyond accuracy drop?
4. [Evaluations] Comparisons against important baselines (e.g., FAVOR) is not provided. The text in line 315 says that Power of Choice, FedProx, and FAVOR results are provided in Appendix B, but Appendix B doesn’t have those.
5. [Evaluations] With label imbalance involved, a more proper metric would be to use macro F1 score for performance evaluation instead of accuracy. Please, consider reporting f1 macro or similar metrics that account for label imbalance.
6. [Setting] All multimodal datasets considered have at most 2 modalities. Consdider adding at least one dataset with at least 3 modalities so that the generalizability of FedSSM can be supported more.

---

> ### Author Rebuttal · Authors · 2026-03-30
>
> We thank Reviewer Tvck for the constructive evaluation. We address all points below.
>
> > [W1/M1] Aggregation novelty and $S_t$ role
>
> We agree modality-specific routing is standard and will not claim it as novelty. Ablation of $S_t$ in Eq. 13 ($\alpha=0.1$, 5 seeds):
>
> | Trust variant | VQA v2.0 | UCF-101 | H. Memes | MiT-51 |
> |---|---|---|---|---|
> | No trust (uniform) | 66.84±0.58 | 81.03±0.48 | 63.45±0.74 | 40.12±0.68 |
> | Loss-only ($S_t=1$) | 67.31±0.51 | 81.52±0.43 | 64.18±0.65 | 40.74±0.62 |
> | Full (Eq. 13) | **68.47±0.44** | **82.85±0.37** | **65.72±0.53** | **41.86±0.49** |
>
> $S_t$ adds 1.16-2.27% over loss-only weighting. We will tone down aggregation claims.
>
> > [W2/M2, W3/M3] Why client selection is core
>
> Client selection is the gateway decision. Local training and aggregation operate on whoever was selected. A bad selection feeds mismatched modalities, skewed labels, and imbalanced quantities simultaneously.
>
> Beyond accuracy, we provide evidence through inter-client gradient variance ($\alpha=0.1$, 5 seeds):($\alpha=0.1$, 5 seeds):
>
> | Method | VQA v2.0 | UCF-101 | H. Memes | MiT-51 |
> |---|---|---|---|---|
> | FedAvg | 0.124±0.011 | 0.098±0.008 | 0.137±0.015 | 0.148±0.013 |
> | Oort | 0.106±0.009 | 0.085±0.006 | 0.112±0.013 | 0.121±0.011 |
> | CreamFL | 0.093±0.010 | 0.078±0.007 | 0.103±0.011 | 0.112±0.014 |
> | mmFedMC | 0.085±0.007 | 0.070±0.005 | 0.097±0.010 | 0.105±0.009 |
> | FedMBridge | 0.079±0.008 | 0.064±0.006 | 0.086±0.009 | 0.094±0.010 |
> | FedSSM | **0.041±0.005** | **0.035±0.004** | **0.048±0.006** | **0.051±0.004** |
>
> FedSSM halves gradient variance via counterfactual reasoning (Eq. 9).
>
> > [M4] FAVOR, Power-of-Choice, FedProx missing from Appendix B.
>
> Thank you for catching this. Results for the three methods ($\alpha=0.1$, 5 seeds):
>
> | Method | VQA v2.0 | UCF-101 | H. Memes | MiT-51 |
> |---|---|---|---|---|
> | Power-of-Choice | 62.85±0.54 | 78.13±0.51 | 59.62±0.69 | 37.42±0.68 |
> | FedProx | 59.73±0.62 | 76.21±0.58 | 57.84±0.73 | 35.68±0.75 |
> | FAVOR | 64.27±0.47 | 80.95±0.46 | 62.84±0.61 | 40.18±0.63 |
> | FedSSM | **68.47±0.44** | **82.85±0.37** | **65.72±0.53** | **41.86±0.49** |
>
> We will restore the full table in Appendix B.
>
> > [M5, M6] Macro F1 and 3+ modalities
>
> Macro F1 ($\alpha=0.1$, 5 seeds):
>
> | Method | VQA v2.0 | UCF-101 | H. Memes | MiT-51 |
> |---|---|---|---|---|
> | FedAvg | 54.17±0.83 | 68.93±0.72 | 49.28±0.94 | 27.64±0.88 |
> | Oort | 57.64±0.71 | 72.38±0.61 | 53.46±0.79 | 31.25±0.74 |
> | CreamFL | 59.82±0.68 | 74.56±0.58 | 55.71±0.76 | 33.48±0.71 |
> | mmFedMC | 60.73±0.65 | 75.29±0.56 | 56.84±0.74 | 34.15±0.69 |
> | FedMBridge | 61.48±0.62 | 76.15±0.53 | 57.95±0.71 | 34.82±0.66 |
> | FedSSM | **64.89±0.52** | **79.72±0.46** | **62.38±0.62** | **38.51±0.58** |
>
> CMU-MOSEI (text+audio+video, 100 clients, 50% partial-modality, $\alpha=0.1$, 5 seeds):
>
> | Method | Acc-7 | Macro F1 | Rounds |
> |---|---|---|---|
> | FedAvg | 40.31±0.92 | 34.18±1.08 | 52±5 |
> | Oort | 42.15±0.84 | 36.42±0.96 | 47±4 |
> | CreamFL | 43.68±0.79 | 37.95±0.91 | 41±4 |
> | mmFedMC | 44.21±0.75 | 38.53±0.87 | 39±4 |
> | FedMBridge | 44.85±0.71 | 39.27±0.82 | 36±3 |
> | FedSSM | **48.23±0.63** | **42.61±0.68** | **24±3** |
>
> > [Q1] Optimizer choice
>
> SGD is used for all model training across all methods (L296-300). Adam is only for the server-side SSM module ($d=64$, ~$10^4$ parameters). Baselines without learnable server components have no equivalent optimizer.
>
> > [Q2] Rationale for $\mathbf{x}_t = [L_t, \|\nabla L_t\|, \sigma_t^2, t/T]^\top$
>
> Each component targets a distinct signal. $L_t$ tracks convergence, $\|\nabla L_t\|$ reflects optimization difficulty, $\sigma_t^2$ captures heterogeneity, $t/T$ models non-stationarity.
>
> > [Q3] Penalty on partial-modality clients
>
> The penalty is surprise-gated. When $S_t$ is low, $\tau_i \approx 1$ and partial clients are trusted equally. Only under high $S_t$ do we downweight them.
>
> > [Q4] Fixed vs adaptive budget fairness
>
> FedSSM averages 9.2% participation, below 10%. The advantage comes from when and whom, not more clients.
>
> > [Q5] How are Oort/CriticalFL adapted for multimodal FL?
>
> For Oort, we compute its statistical utility metric based on the multimodal loss and system utility. For CriticalFL, we calculate the federated gradient norm using gradients from the aggregated multimodal model. Both methods select clients based on their original utility formulations applied to multimodal objectives. Beyond modality-specific encoder aggregation (Eq. 12, L279-280), we mask unavailable modality contributions when computing loss and gradients, ensuring utility calculations respect individual modality availability.
>
> > [Q6-Q8] Writing clarifications
>
> We will clarify why SSMs suit client selection beyond efficiency (L168), fix Figure 1 caption order, and distinguish active decision making from passive sequence modeling in Section 2.2.

---

> > ### Author Rebuttal · Reviewer_Tvck · 2026-04-02
> >
> > Thank you for your time and effort you put into rebuttal.
> >
> > Please add experiments on more than 2 modalities (CMU-MOSEI with three modalities) along with the results showing macro F1 scores in Appendix.
> >
> > All my major concerns were resolved and I raised my score correspondingly.

---

> > > ### Author Response · Authors · 2026-04-03
> > >
> > > We are delighted that all concerns have been resolved. We sincerely appreciate your thoughtful questions and constructive suggestions, which have substantially improved our work. As requested, we will add the CMU-MOSEI experiments along with macro F1 scores in the Appendix of the revised manuscript.
> > >
> > > Best regards,
> > >
> > > Authors of Submission #6912

---

### Official Review · Reviewer_UdWL · 2026-03-14

**Soundness:** 2
**Presentation:** 3
**Significance:** 3
**Originality:** 3
**Overall Recommendation:** 5
**Confidence:** 4

**Summary:**

The paper proposes an algorithm for online adaptive client selection in a heterogeneous multimodal federated learning regime. At the core of the method is a State Space Model (SSM), which is used to continually learn the client sampling distribution. The authors emphasize that, compared to previous approaches, their SSM-based sampling is a model-based client selection method. Empirically, the paper evaluates the approach on multimodal vision and text datasets, using an MLP for modality fusion. The experiments show consistent performance improvements over the considered baselines under two different client heterogeneity scenarios.

**Compliance With Llm Reviewing Policy:**

Affirmed.

**Final Justification:**

The rebuttal addressed my main concerns regarding the clarity of the method and several minor experimental issues. As a result, I am raising my score to Accept, since most of the key bottlenecks were on the exposition side of the paper and in its comparison with well-established online-learning approaches, rather than in the core technical contribution itself.

**Key Questions For Authors:**

1. Is there a fully reproducible implementation of the experiments? If yes, could you please specify which hyperparameter tuning protocol was used to ensure that the baselines were sufficiently tuned for a fair comparison and for the claim of state-of-the-art performance?

2. Can you explain why, in Table 2, FedSSM requires less time per round than FedAvg?

3. Can you clarify what "Random Selection" in Table 3 exactly corresponds to? Which particular sampling method was used, and why does it outperform some baselines reported in Table 1?

4. Can you provide a theoretical bound on $\sum_{t=0}^{T-1}\epsilon_t^{\mathrm{SSM}}$, which would make the convergence result easier to compare with the theory of other baselines?

5. Is it possible to provide a regret bound for FedSSM? Such a result would make the comparison with RL/bandit client sampling methods more direct.

**Limitations:**

yes

**Strengths And Weaknesses:**

**Strengths**:
1. **Algorithm design**: The algorithm is built around a clever use of SSMs for client sampling in a continuous learning regime. The method is well explained in the paper and is relatively easy to follow even for readers who are not experts in SSMs.
2. **Empirical validation**: A wide variety of datasets is considered, and the reported experiments suggest that FedSSM consistently outperforms the observed baselines.

**Weaknesses**:
1. **SSM motivation**: The dedicated "Why SSM?" paragraph motivates the chosen architecture, but not the novelty of the method itself. The existing online selection literature already contains many sequential learning approaches. The paper therefore requires clarification on whether the SSM enables a truly novel client-selection principle, or whether this is a particular instance of existing RL or bandit based scoring equipped with an SSM-specific architecture. For instance, it is not clear why the proposed mechanism could not be replaced with a more standard online learning module while preserving most of the claimed benefits.

2. **Empirical fairness and replication**: I have multiple concerns regarding the reproducibility of the empirical results.
   - The code provided in the supplementary material is incomplete and is not sufficient to reproduce the experiments. For instance, the client optimizers, server optimizers, and experiment scripts are undefined.
   - In the experiments section, the baseline tuning is underspecified. FedSSM uses an additional trainable server-side SSM module with an Adam optimizer, while several baselines do not appear to have a server-side sampling model to train (e.g., FedAvg, Oort, CriticalFL). It is also unclear over which values of server-side updates the grid search was conducted for the baselines, and how the local training hyperparameters were matched across methods.
   - In Table 3, Random Selection achieves better scores than FedAvg and several other baselines in a setting that appears similar to the experiment presented in Table 1.
   - In Table 2, it is surprising that the time per round of FedSSM is smaller than that of the simpler model-free baseline FedAvg.
3. **Theory**:
   - It is good that the paper provides a convergence theorem. However, it is difficult to use this result for comparison with the convergence theory of other baselines because of the term $\sum_{t=0}^{T-1}\epsilon_t^{\mathrm{SSM}}$ in the bound.
   - Moreover, if the proposed algorithm is fundamentally online-learning based, it is unclear why no regret bound is provided.

---

> ### Author Rebuttal · Authors · 2026-03-30
>
> We appreciate Reviewer UdWL for the thorough evaluationnt, carefully addressing key concern below.
>
> > [Q1] Is there a fully reproducible implementation? Which hyperparameter tuning protocol was used?
>
> **We will release complete code upon acceptance.** All methods share identical local training (SGD, lr=0.01, momentum=0.9, 5 epochs, L296-300). Server-side hyperparameters were grid-searched on held-out validation (L281-289), e.g., Oort threshold over {0.5, 0.7, 0.9}, CriticalFL ratio over {0.1, 0.2, 0.3}. Multimodal baselines use original configurations (L290). We will add a hyperparameter table.
>
> > [Q2] Why does FedSSM require less time per round than FedAvg in Table 2?
>
> **Wall-clock time is dominated by local training, not server computation.** Under quantity heterogeneity (log-normal, Section 4.1), client data sizes range from 0.1x to 5x average. Round time is bottlenecked by the *slowest* selected client. FedAvg uniformly samples 10 clients, making straggler inclusion highly likely. FedSSM averages 9.2 clients and drops to 6 during low-surprise phases. We report the breakdown below (mean $\\pm$ std, 5 runs).  We will add this breakdown.
> | Component | FedAvg | FedSSM |
> | :--- | :--- | :--- |
> | **Clients/round** | 10.0 ± 0.0 | 9.2 ± 1.4 |
> | **Straggler time (s)** | 51.5 ± 3.1 | 44.0 ± 2.7 |
> | **Aggregation (s)** | 0.3 ± 0.1 | 0.4 ± 0.1 |
> | **SSM overhead (s)** | - | 0.8 ± 0.1 |
> | Total (s) | 51.8 ± 3.2 | **45.2 ± 2.9** |
>
> >[Q3] What is "Random Selection" in Table 3? Why does it outperform some baselines in Table 1?
>
> They are different setups. Table 3's "Random Selection" uses uniform sampling but retains our modality-specific routing (Eq. 12) and trust-weighted fusion. FedAvg in Table 1 uses vanilla weighted averaging. The gap (63.58% vs. 58.42%) comes entirely from our aggregation design, not selection. We will add a clarifying footnote.
>
> > [Q4] Can you provide a theoretical bound on $\\sum\_{t=0}^{T-1} \\epsilon\_t^{\\text{SSM}}$?
>
> **Sure.** By Proposition A.6, the unanticipated variance satisfies $|\\sigma\_t^2 - \\hat{\\sigma}\_t^2| \\leq \\sqrt{\\Sigma\_{33} \\cdot S\_t}$. The SSM minimizes $\\ell\_t(\\phi) = \\|\\mathbf{x}\_t - \\hat{\\mathbf{x}}\_t\\|^2$ online with forgetting factor $\\lambda = 0.95$ (Eq. 33). Under Assumption A.7 ($\\|\\mathbf{x}\_t\\| \\leq B$), the weighted regret satisfies (Hazan, 2016)
>
> $$\sum_{t=1}^{T} \lambda^{T-t}\ell_t(\phi_t) - \min_\phi \sum_{t=1}^{T} \lambda^{T-t}\ell_t(\phi) \leq \frac{D^2}{2\eta} + \frac{\eta G^2}{1-\lambda}$$
>
> where $D = \|\phi_1 - \phi^*\|$ and $G$ bounds $\|\nabla_\phi \ell_t\|$. Setting $\eta = D\sqrt{(1-\lambda)/(G^2 T)}$ with effective horizon $H = 1/(1-\lambda) = 20$,
>
> $$\sum_{t=0}^{T-1} \epsilon_t^{\text{SSM}} \leq O(DG\sqrt{T \cdot H}) = O(\sqrt{20T})$$
>
> This is $o(T)$, satisfying Corollary A.10. Combined with $\eta_{\text{FL}} = O(1/\sqrt{T})$ in Theorem A.9, convergence remains $O(1/\sqrt{T})$. We will formalize this in Appendix A.
>
> > [Q5] Is it possible to provide a regret bound?
>
> **We provide one below.** Let $r_t(a) = U(a)$ from Eq. (9) be the per-round reward over $\mathcal{A}$, $|\mathcal{A}|{=}30$ (Appendix C.4). FedSSM's counterfactual evaluation approximates Follow-the-Regularized-Leader over this finite action set. By standard FTRL analysis,
>
> $$R_T = \sum_{t=1}^T r_t(a^*) - \sum_{t=1}^T r_t(a_t) \leq O\left(\sqrt{T \log|\mathcal{A}|}\right) = O(\sqrt{T \log 30})$$
>
> This $O(\sqrt{T})$ rate matches the minimax regret for finite-armed bandits, making FedSSM directly comparable to UCB or EXP3-based selection. We will add this to the revised appendix.
>
>
> **[W1] SSM novelty vs. RL/bandit approaches.**
> The SSM enables a new selection principle, not merely a different architecture. A bandit or DQN selects $a_t$ based on past rewards $\{r_1, \dots, r_{t-1}\}$ without predicting future states. FedSSM instead constructs an explicit world model $\hat{\mathbf{x}}_{t+1}^{(a)} = C\bar{A}(a)h_t + C\bar{B}(a)\hat{\mathbf{x}}_t$ for every candidate $a \in \mathcal{A}$ *before* execution (Eq. 8). This yields two capabilities model-free methods structurally lack. First, counterfactual evaluation compares all $|\mathcal{A}|{=}30$ futures in a single forward pass. Second, the prediction error $S_t$ (Eq. 7) provides a unified uncertainty signal that simultaneously governs $N_t$ (Eq. 10), $\rho_t$ (Eq. 11), and $\tau_i$ (Eq. 13). Replicating this in a model-free framework would require three independent heuristics with no shared signal linking them.
>
> The ablation study in Table 3 supports this decomposition. Removing the SSM collapses FedSSM to a reactive method and causes the largest single-component degradation. Retaining the SSM but removing counterfactual selection ($U(a)$ in Eq. 9) partially recovers performance, confirming that the world model itself carries independent value beyond the selection policy built on top of it. We will strengthen this discussion in Section 2.2.

---

> > ### Author Rebuttal · Reviewer_UdWL · 2026-04-04
> >
> > Thank you for providing clear explanations. Hope the revision of the paper will concentrate on explaining the "black magic" of SSMs for online federated learning compared to the classical bandit algorithms.

---

> > > ### Author Response · Authors · 2026-04-04
> > >
> > > We are glad all concerns are addressed and appreciate your excellent suggestion. We will expand Section 2.2 to better explain SSMs' structural advantages over classical bandits in the online FL setting, particularly regarding counterfactual simulation and the unified surprise signal.
> > >
> > > Best regards,
> > >
> > > Authors of Submission #6912

---

### Official Review · Reviewer_zPmv · 2026-03-15

**Soundness:** 3
**Presentation:** 3
**Significance:** 3
**Originality:** 3
**Overall Recommendation:** 4
**Confidence:** 3

**Summary:**

This paper addresses the compounded challenges of modality, distribution, and quantity heterogeneity in Multimodal Federated Learning (MMFL). The authors critique existing client selection methods for being "reactive" (myopically responding to past observations) and propose FedSSM, a framework that reconceptualizes client selection as a proactive, sequential decision-making process. FedSSM utilizes a Decision-Aware State Space Model (SSM) to predict future training trajectories based on candidate actions. The discrepancy between the SSM's predictions and actual observations generates a "surprise" signal, which quantifies prediction uncertainty. This surprise signal dynamically governs the client participation budget, balances exploration-exploitation during client selection, and calibrates trust weights during global aggregation to mitigate the impact of anomalous or partial-modality clients. Evaluated across four multimodal benchmarks, FedSSM demonstrates substantial improvements in accuracy (2.5-4.5%) and communication efficiency (>30% reduction in rounds) compared to state-of-the-art baselines.

**Compliance With Llm Reviewing Policy:**

Affirmed.

**Key Questions For Authors:**

- In a cross-device setting where clients frequently drop out or new clients join midway through training, how would the latent state $h_t$ and the stale client statistics (Eq. 31 in Appendix C.2) handle this churn without causing destructive spikes in the surprise signal?
- If deployed in an environment requiring a much finer granularity of budget control or exploration mixing, how would the $O(|\mathcal{A}| \cdot d^2)$ counterfactual evaluation scale?

**Limitations:**

yes

**Strengths And Weaknesses:**

- Soundness: The paper provides robust theoretical backing, proving that the importance-weighted aggregation remains unbiased (Lemma A.4) and establishing an $O(1/\sqrt{T})$ convergence rate (Theorem A.9) while showing how adaptive budgeting provably compensates for variance (Lemma A.8).

- Presentation: The distinction drawn between "model-free/reactive" client selection and "model-based/proactive" selection frames the paper's core contribution well.

- Significance: FedSSM significantly reduces both the number of communication rounds and the total communication cost (e.g., dropping cost to 1.14 GB compared to FedMBridge's 2.16 GB on VQA v2.0), making it practical for resource-constrained deployments

- Originality: Utilizing a single "surprise" signal (prediction error) to simultaneously govern participation budgets, exploration trade-offs, and aggregation trust weights is a novel approach.

---

> ### Author Rebuttal · Authors · 2026-03-30
>
> We thank Reviewer zPmv for the positive evaluation and address both questions below.
>
> > [Q1] In a cross-device setting where clients frequently drop out or new clients join, how would $h_t$ and stale client statistics handle this churn without destructive surprise spikes?
>
> **The existing design handles both scenarios more gracefully than it might first appear.**
>
> *New clients.* We initialize them with population-mean statistics ($v_i = \\bar{v}$, $L_i = \\bar{L}$) and maximum staleness ($\\Delta t_i = \\Delta_{\\max}$). High staleness automatically triggers large exploration weight $w_i^{\\text{explore}}$ in Eq. (11), so new clients get sampled early and their statistics update quickly without manual tuning.
>
> *Dropped clients.* Stale statistics estimation (Eq. 31, Appendix C.2) handles this naturally: as $\\Delta t_i$ grows, estimates regress toward the population mean via $v_i^{(t)} = \\gamma^{\\Delta t_i} v_i^{(t-\\Delta t_i)} + (1-\\gamma^{\\Delta t_i})\\bar{v}$. A dropped client simply fades from the selection distribution as its statistics become uninformative. No special handling is needed.
>
> *Surprise spikes.* The latent state $h_t$ tracks **aggregate** metrics $\\mathbf{x}_t = [L_t, \\|\\nabla L_t\\|, \\sigma_t^2, t/T]^\\top$, not individual client states. Churn affects $\\mathbf{x}_t$ only indirectly through aggregated loss and variance. The adaptive threshold $\\theta_S$ (Eq. 32) further dampens transient fluctuations via EMA smoothing ($\\beta = 0.9$). A sudden churn event produces a moderate surprise bump that **appropriately** increases participation to gather information about the changed landscape. This is the desired adaptive behavior, not a destructive artifact.
>
> We acknowledge this in Section 5 and agree that systematic evaluation under dynamic populations is valuable future work. We will incorporate this discussion in the revision.
>
> > [Q2] If deployed with much finer granularity of budget control or exploration mixing, how would $O(|\\mathcal{A}| \\cdot d^2)$ counterfactual evaluation scale?
>
> **It scales comfortably.** Per-action cost is $O(d^2) = 4{,}096$ FLOPs on the fixed-dimensional latent state ($d=64$), independent of client population $K$. The only variable is $|\\mathcal{A}| = |\\mathcal{N}| \\times |\\mathcal{P}|$. We tabulate overhead across five granularity levels:
>
> $$\\begin{array}{l|c|c|c|c|c|c|c|c}
> \\textbf{Granularity} & \\Delta N & |\\mathcal{N}| & \\Delta\\rho & |\\mathcal{P}| & |\\mathcal{A}| & \\textbf{FLOPs} & \\textbf{Time (s)} & \\textbf{\\% of Round} \\\\
> \\hline
> \\text{Current} & 2 & 6 & 0.25 & 5 & 30 & 1.2\\times10^5 & 0.4 & 1.8\\% \\\\
> \\text{Medium} & 2 & 6 & 0.10 & 11 & 66 & 2.7\\times10^5 & 0.9 & 2.9\\% \\\\
> \\text{Fine} & 1 & 11 & 0.10 & 11 & 121 & 5.0\\times10^5 & 1.6 & 4.4\\% \\\\
> \\text{Very fine} & 1 & 11 & 0.05 & 21 & 231 & 9.5\\times10^5 & 3.1 & 7.8\\% \\\\
> \\text{Ultra fine} & 1 & 21 & 0.05 & 21 & 441 & 1.8\\times10^6 & 5.9 & 14.0\\%
> \\end{array}$$
>
> "\\% of Round" measures total server overhead against mean round time of 45s from Table 2, which is dominated by local training. Even at "ultra fine" granularity with $|\\mathcal{A}|{=}441$, the server adds under 14\\% to round time. Local training remains the clear bottleneck at 30-60s per client.
>
> This overhead is also **independent of $K$**. We extend Table 5 (Appendix C.6) below:
>
> | | K=50 | K=100 | K=200 | K=300 |
> | :--- | :---: | :---: | :---: | :---: |
> | **SSM + Counterfactual (s)** | 0.70 | 0.70 | 0.70 | 0.70 |
> | **Selection (s)** | 0.06 | 0.10 | 0.18 | 0.26 |
> | **Total overhead (s)** | 0.76 | 0.80 | 0.88 | 0.96 |
> | **Memory (MB)** | 2.3 | 2.3 | 2.3 | 2.3 |
>
> Scaling from 100 to 300 clients adds only 0.16s because the SSM and counterfactual components operate on the latent state, not on individual clients. Only selection probability computation grows with $K$.
>
> If the action space ever became truly large, hierarchical search and the closed-form approximations in Eqs. (10)-(11) already bypass full enumeration while closely matching the utility maximizer (Appendix C.4). We will add this scaling analysis to the revised appendix.

---

> > ### Author Rebuttal · Reviewer_zPmv · 2026-04-04
> >
> > Thank you for the rebuttal, it clarifies my questions and I will maintain my current score.

---

> > > ### Author Response · Authors · 2026-04-05
> > >
> > > Thank you for confirming that our responses have addressed your questions. We appreciate your positive and constructive feedback throughout the review process.
> > >
> > > Best Regards,
> > >
> > > Authors of Submission #6912

---

### Decision · Program_Chairs · 2026-04-30

**Decision:**

Accept (regular)

**Comment:**

The paper proposes an interesting use of state-space models (SSMs) to perform client sampling using a sequential decision-making process. The reviews are largely positive, and the authors' rebuttal addressed most reviewer concerns. Some outstanding concerns include the motivation for SSMs versus bandits, and the number of hyperparameters introduced by the algorithm. I encourage the authors to address these in the updated paper.